# Scleral PERK and ATF6 as targets of myopic axial elongation of mouse eyes

Shin-ichi Ikeda [1,2], Toshihide Kurihara [1,2] ✉, Xiaoyan Jiang[1,2], Yukihiro Miwa [1,2], Deokho Lee [1,2], Naho Serizawa[1,2], Heonuk Jeong[1,2], Kiwako Mori[1,2], Yusaku Katada[1,2], Hiromitsu Kunimi[1,2], Nobuhiro Ozawa[1,2], Chiho Shoda[1,2], Mari Ibuki[1,2], Kazuno Negishi [2], Hidemasa Torii [1,2] & Kazuo Tsubota [2,3] ✉

Axial length is the primary determinant of eye size, and it is elongated in myopia. However, the underlying mechanism of the onset and progression of axial elongation remain unclear. Here, we show that endoplasmic reticulum (ER) stress in sclera is an essential regulator of axial elongation in myopia development through activation of both PERK and ATF6 axis followed by scleral collagen remodeling. Mice with lens-induced myopia (LIM) showed ER stress in sclera. Pharmacological interventions for ER stress could induce or inhibit myopia progression. LIM activated all IRE1, PERK and ATF6 axis, and pharmacological inhibition of both PERK and ATF6 suppressed myopia progression, which was confirmed by knocking down above two genes via CRISPR/ Cas9 system. LIM dramatically changed the expression of scleral collagen genes responsible for ER stress. Furthermore, collagen fiber thinning and expression of dysregulated collagens in LIM were ameliorated by 4-PBA administration. We demonstrate that scleral ER stress and PERK/ATF6 pathway controls axial elongation during the myopia development in vivo model and 4-PBA eye drop is promising drug for myopia suppression/treatment.

Myopia has been growing at a pandemic rate[1]. The prevalence of visual impairment or retinal implication rose with increasing axial length, which is the primary pathological change of myopia[2,3]. Thus, there are emerging needs to elucidate underlying pathogenesis of axial elongation and to develop effectively and safely preventive and therapeutic intervention[4]. Axial elongation is concomitant with scleral extracellular matrix (ECM) remodeling, including changes in ECM genes expressions and collagen fiber thinning[5,6]. However, the underlying mechanism of the onset and progression of ECM remodeling and axial elongation remains unclear.

Organ size is controlled by external and internal stimuli and is one of the important determinants of organ function. During changes in the organ size, such as in developmental/growth stages or pathological conditions, endoplasmic reticulum (ER) stress, and/or series of ER

stress-sensing proteins such as inositol-requiring enzyme 1 (IRE1), protein kinase RNA-like ER kinase (PERK), and activating-transcription factor 6 (ATF6), may contribute to these changes by disturbing gene expression and translation, inducing apoptosis and/or collagen deposition, followed by interstitial fibrosis[7–14]. We hypothesized that scleral ER stress participates in the onset and progression of axial elongation during the myopia development as axial elongation is a change in the organ size, which is probably achieved by remodeling of ECM in sclera. In fact, there have been recent reports indicating that ER stress occurs in the myopic sclera using a form-deprivation myopia model in guinea pig[15]. To elucidate the hypothesis, we used pharmacological and genetic methods to induce/inhibit ER stress in a mouse model of LIM and tested its effect on the myopia progression. Here, we show that scleral ER stress is essential for myopia development and

[1]Laboratory of Photobiology, Keio University School of Medicine, 35 Shinanomachi, Shinjuku-ku, Tokyo 160-8582, Japan. [2]Department of Ophthalmology, Keio University School of Medicine, 35 Shinanomachi, Shinjuku-ku, Tokyo 160-8582, Japan. [3]Tsubota Laboratory, Inc., 34 Shinanomachi, Shinjuku-ku, Tokyo 160-0016, Japan. ✉e-mail: kurihara@z8.keio.jp; tsubota@tsubota-lab.com

intervention of scleral ER stress is promising strategy for myopia prevention/treatment.

## Results

### ER stress occurs in the sclera of minus lens-induced myopia

The eyes of a murine LIM model (Supplemental Fig. 1a) showed axial elongation (Fig. 1a), myopic shift in refraction (Fig. 1b), attenuation of VCD shortening during eye growth (Supplemental Fig. 1b) and scleral thinning (Supplemental Fig. 1c–e) compared to the control eyes, which are typical features of myopia in humans. Transmission electron microscopic observation revealed that the rough ER was dilated in scleral fibroblasts in LIM mice (Fig. 1c), indicating ER stress in scleral fibroblasts. Under ER stress, ER membrane sensor proteins such as IRE1, PERK, and ATF6 are activated, followed by the enhancement of transcriptional reprogramming to overcome the stress condition, termed the unfolding protein response (UPR). UPR triggers increase in (1) ER chaperones such as glucose-regulated protein 78 (GRP78; also called BiP), GRP94; (2) ER-associated protein degradation (ERAD)-related molecules such as ER degradation-enhancing α-mannosidase-like protein (EDEM) and ER-localized DnaJ (ERdj3, 4, and 5); (3) molecules controlling the translational rate, such as growth arrest and DNA damage-inducible protein (GADD34), suppressor/enhancer of Lin-12-like (SEL1L); and (4) transcriptional factors such as ATF4 and C/EBP homologous protein (CHOP)[16,17]. LIM increased the phosphorylation level of IRE1 and PERK, the ratio of cleaved activating ATF6 (ATF6-N) per full length of ATF6 (ATF6-P) and the phosphorylation level of eukaryotic translation initiation factor 2α (eIF2α) which is downstream of PERK in the sclera (Fig. 1d, e). ER stress marker genes expression was upregulated in myopic sclerae compared to those in control sclerae 3 weeks after LIM (Fig. 1f). Since there were no such differences between the non-lens wearing eyes and the age-matched control eyes, it can be assumed that there is no stress caused by surgery procedure or wearing frame without lens (Fig. 1d–f). Immunohistochemistry using the anti-BiP (GRP78) antibody revealed that the number of BiP-positive cells were higher in LIM sclerae compared to that in control sclerae (Supplemental Fig. 2a, b). Rough ER dilation and increased expression in ER stress marker genes such as *BiP* and *CHOP* were also observed in the sclera of a chick LIM model as well (Supplemental Fig. 1f, g). Thus, myopic sclerae are under ER stress by lens-induced defocus across species. On the other hand, in the retinal layer, there was no LIM-induced activation of IRE1, eIF2, ATF6, or increased expression of their downstream genes (Supplemental Fig. 2e, f). ER of the retinal pigment epithelium (RPE) was observed by TEM, but no expansion of the ER was observed (Supplemental Fig. 2g), suggesting that LIM does not cause ER stress in the retinal layer.

### Intervention in scleral ER stress can control myopia progression

To determine whether scleral ER stress is associated with axial elongation and myopia development, 4-phenylbutyric acid (4-PBA), which acts as a chemical chaperone attenuating ER stress, was intraperitoneally injected during LIM, and myopic changes were compared between PBS- and 4-PBA-injected mice. LIM activated IRE1, PERK (evaluated by eIF2 phosphorylation) and ATF6 pathways (Fig. 2a, b) and increased ER stress marker gene expression (Supplemental Fig. 3a) in PBS-injected mouse sclerae. In contrast, the increases were diminished by 4-PBA injection (Fig. 2a, b and Supplemental Fig. 3a). Concomitant with ER stress attenuation, myopic shift in refraction and axial elongation was diminished in 4-PBA-injected mice both for 1 week LIM (Supplemental Fig. 3b, c) and 3 weeks LIM (Fig. 2c, d). As the eye grows, sum of the length of the vitreous chamber depth (VCD) and retinal thickness (RT) shortens in mouse eyes, and myopia induction attenuates this shortening change[18], but 4-PBA also normalized the VCD + RT change (Supplemental Fig. 3d). Administration of 4-PBA did not affect the lens thickness (Supplemental Fig. 4j). The administration of another chemical chaperone, tauroursodeoxycholic acid (TUDCA),

also inhibited LIM-induced myopic changes in refraction, axial elongation and attenuation of VCD + RT shortening (Supplemental Fig. 3e–g). To answer the question of whether systemically administered 4-PBA reaches the eye and even the sclera and produces an inhibitory effect on the myopia progression, we measured the amount of 4-PBA and its metabolite, phenylacetic acid (PAA), in the ocular tissues (retina, choroid, and sclera) of 4-PBA-treated (1 week) mice by LC-MS/MS. Compared to the retina and the choroid, the sclera had the highest amount of 4-PBA detected (Supplemental Fig. 9 and Supplemental Table. 4). On the other hand, PAA, a metabolite produced when 4-PBA undergoes β-oxidation, was most abundant in the retina, followed by the choroid and sclera (Supplemental Fig. 10 and Supplemental Table. 5). These results suggest that systemically administered 4-PBA reaches the sclera and continues to exist without undergoing metabolism. Since a high amount of PAA was detected in the retina, we thought that it might have a negative effect on retinal function, so we measured ERG after 1 week 4-PBA administration and evaluated the effect on retinal function. Intraperitoneal administration of 4-PBA (200 mg/kg/day) decreased the amplitudes of both scotopic a-wave and scotopic b-wave, as well as photopic b-wave (Supplemental Fig. 4a–d). These results suggest that systemic administration of 4-PBA may be harmful to the retina.

As topical eyedrops are generally used to treat eye diseases, we next examined whether the instillation of 4-PBA could effectively suppress the myopia progression. 4-PBA instillation (2%) supressed activations of LIM-induced IRE1, PERK (determined by eIF2) and ATF6 pathways (Fig. 2e, f) and dose-dependently inhibited axial elongation and refractive error (Fig. 2g, h). Instillation of 4-PBA also normalized VCD + RT change (Supplemental Fig. 3h). It was confirmed that the eye-dropped 4-PBA also reached the sclera (Supplemental Tables 6 and 7). Since systemic administration of 4-PBA had an adverse effect on retinal function, ERG was also measured after 1-week ocular administration of 4-PBA. There was no effect of 4-PBA instillation on ERG amplitude (Supplemental Fig. 4e-h). Furthermore, to verify the effect of 4-PBA eye drop on the anterior segment of the eye, we measured mucin expression in the conjunctiva and the thickness of the lens as ER stress and/or 4-PBA associates with mucin production and lens homeostasis[19,20]. Instillation of 4-PBA for 3-week did not affect the expression level of mucin 1, 4 and 5ac (Supplemental Fig. 4i). In addition, ocular administration of 4-PBA tended to reduce the thickness of the lens (Supplemental Fig. 4k). Lens thickening due to minus lens wearing is thought to be a compensatory effect for focus shift, but since myopia was suppressed by 4-PBA eye drops, lens thickening due to minus lens wearing did not occur in the 4-PBA instillation group.

To evaluate whether 4-PBA administration is also effective for the myopia treatment, topical administration of 4-PBA started after 3 weeks LIM period (Supplemental Fig. 5a). Myopic phenotypes were observed in the vehicle-instilled eyes 1 and 3 weeks after minus lens removal; however, 4-PBA-instilled eyes showed a decreased myopic phenotype, suppressed axial elongation (Supplemental Fig. 5b–f), and a hyperopic shift (Supplemental Fig. 5g–k). These data indicate that interventions to control ER stress in the sclera are effective in suppressing and possibly even treating myopia.

Instillation of tunicamycin (50 μg/ml), an antibiotic synthesized from *Streptomyces chartreusis*, induced ER stress in sclerae 1–24 h after administration, as evaluated by western blotting of IRE1, eIF2, ATF6 and qPCR for CHOP expression (Fig. 3a, b). We measured axial length, refraction and VCD + RT before and 1 week after tunicamycin (or PBS as a control) ocular administration and calculated those changes. A single bout of tunicamycin instillation in 3-week-old C57BL6J mice induced axial elongation, myopic shift in refraction and attenuation of VCD + RT shortening (Fig. 3c, d and Supplemental Fig. 3i). Ocular administration of thapsigargin (10 μM), another ER stress inducer, also induced those myopic changes (Supplemental Fig. 3j–l). Myopic changes induced by tunicamycin or thapsigargin were not observed in

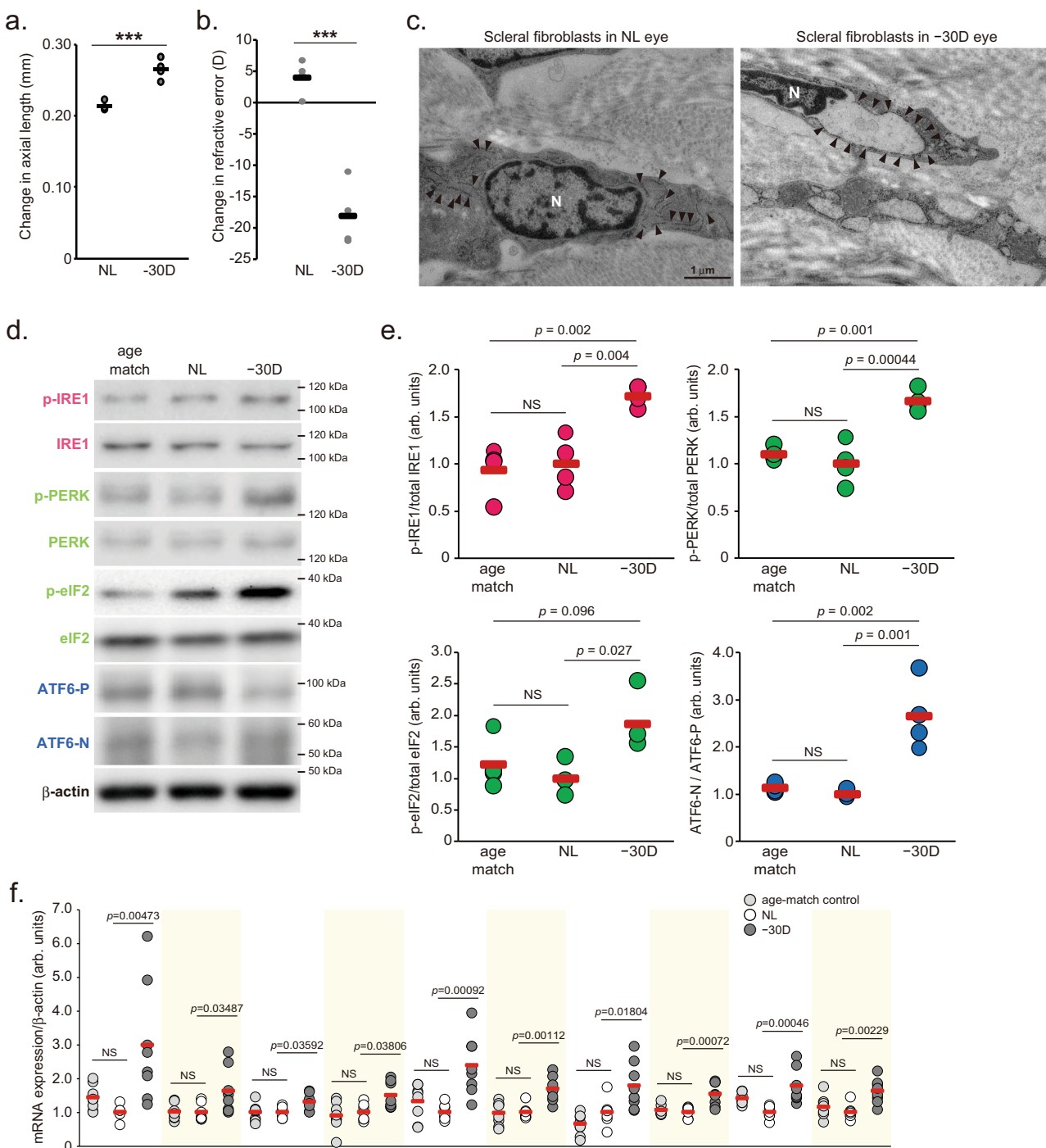

**Fig. 1 | Lens-induced myopia induces axial elongation and endoplasmic reticulum stress in the sclera. a** Changes in axial length during 3-week lens-induced myopia (LIM) in C57BL6J mice ($n = 4$). NL: No Lens control, −30D: minus 30 D lens-wearing eye. ***$p = 0.00062$; Student's two-tailed $t$-test. **b** Changes in refraction during 3-week LIM in C57BL6J mice ($n = 4$); ***$p = 0.00029$; Student's two-tailed $t$-test. **c** Transmission electron microscopy images of no lens (NL)-wearing (left panels) and minus 30 D lens-wearing (right panels) sclerae. Representative images of three biologically independent samples; N indicates nucleus and allow heads indicate rough endoplasmic reticulum. Scale bars: 1.0 μm. **d** Immunoblots showing ER sensor protein activation evaluated by phosphorylation levels of IRE1, PERK, eIF2α (downstream of PERK), and the ATF6 precursor (ATF6-P) and cleaved

form of ATF6 (ATF6-N). Representative blots from four independent experiments are shown. Age-match: age-matched non-treated control, NL: No Lens-wearing eye, −30D: minus 30 D lens-wearing eye. **e** Densitometric quantification of blots in Fig. 1d using ImageJ. NL group was assigned a value of 1.0. Each group were $n = 4$ (each sample pooled 3 sclerae). The $p$ values were determined by one-way ANOVA with Tukey HSD (two-tailed). NS: Not Significant. **f** UPR target gene expression level determined by quantitative PCR in age-matched control (right gray), No-Lens control (white) and LIM (gray) sclerae for 3-week LIM ($n = 7$ or 8). NL group was assigned a value of 1.0. The $p$ values were determined by one-way ANOVA with Tukey HSD (two-tailed). Source data are provided as a Source Data file.

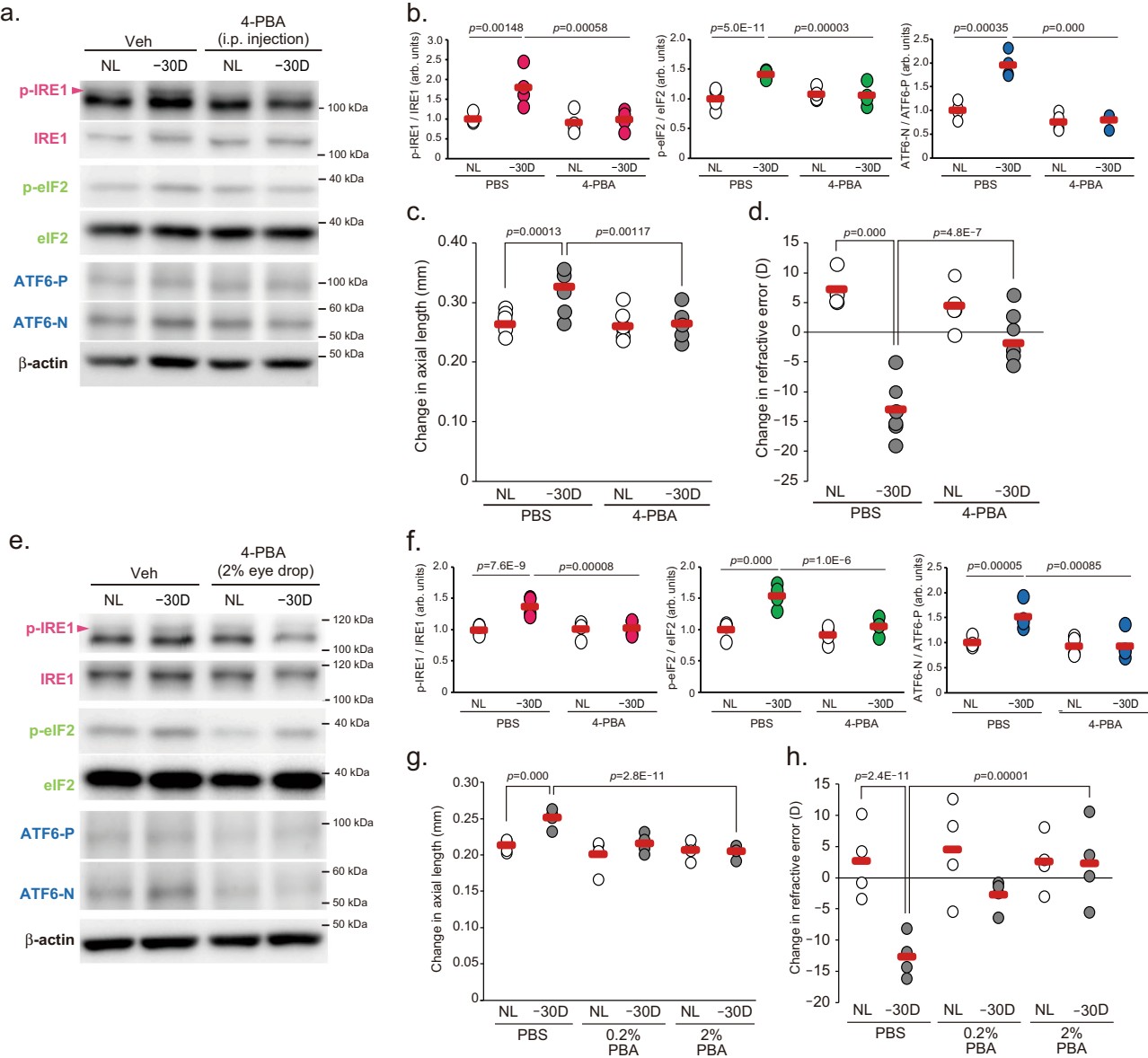

**Fig. 2 | Attenuation of scleral ER stress is sufficient to suppress myopia development. a** Intraperitoneal (i.p.) injection of 4-phenylbutyric acid (4-PBA; 200 mg/kg/day) suppresses Lens-induced myopia (LIM)-induced activation in UPR branch determined by Western blotting in sclerae. NL: No Lens control, −30D: minus 30 D lens-wearing eye. **b** Densitometric quantification of blots in Fig. 2a using ImageJ. PBS-NL group was assigned a value of 1.0. Each group were *n* = 4 (each sample pooled 3 sclerae). The *p* values were determined by two-tailed Generalized Estimating Equations. **c** LIM-induced axial elongation is inhibited by 4-PBA administration for 3 weeks (*n* = 6 per group); **p* < 0.01 (*p* = 0.001), ****p* < 0.001; two-tailed Generalized Estimating Equations. **d** LIM-induced myopic shift in refraction is inhibited by 4-PBA administration for 3 weeks (*n* = 6 per group);

****p* < 0.001; two-tailed Generalized Estimating Equations. **e** Eye drop administration of 4-PBA (2% solution in PBS) during LIM period (to both eyes) suppresses LIM-induced activation in UPR branch determined by Western blotting in sclerae. **f** Densitometric quantification of blots in Fig. 2e using ImageJ.PBS-NL group was assigned a value of 1.0. Each group were *n* = 4 (each sample pooled 3 sclerae). ****p* < 0.001; two-tailed Generalized Estimating Equations. **g** 4-PBA instillation suppresses axial elongation in a dose-dependent manner (*n* = 4 per group); ****p* < 0.001; two-tailed Generalized Estimating Equations. **h** 4-PBA instillation suppresses myopic shift in refraction in a dose-dependent manner (*n* = 4 per group); ****p* < 0.001; two-tailed Generalized estimating equations. Source data are provided as a Source Data file.

the group that received 4-PBA instillation after tunicamycin or thapsigargin treatment (Supplemental Fig. 3m, n). Furthermore, tunicamycin instillation in 5-day-old white Leghorn chicks increased the ocular axial length compared to the PBS-treated group (Supplemental Fig. 3o, p). Together, these results demonstrated that scleral ER stress is an essential mechanism of axial elongation during myopia development.

**PERK and ATF6 pathways are involved in myopia progression**
ER stress is sensed by three trans-ER membrane proteins, IRE1, PERK and ATF6, activating transcriptional programs to reduce the protein-

folding load in the ER. To determine which molecule is responsible for the myopia development, inhibitors of these molecules were instilled during the 10 days LIM period. Ocular administration of STF083010 (STF: IRE1 inhibitor), GSK2656157 (GSK: PERK inhibitor) and nelfinavir (NFV: site-2 proteinase inhibitor that causes ATF6 suppression) suppressed activation of each pathway by LIM in the sclera, although there was some compensatory response (Fig. 4a, b). The axial length, refraction and VCD + RT shortening were comparable between the DMSO (as a control)-instilled and STF-instilled mice in both LIM and non-LIM eyes (Fig. 4c, d and Supplemental Fig. 6c). All the above 3 parameters showed myopic changes when compared to NL and −30D

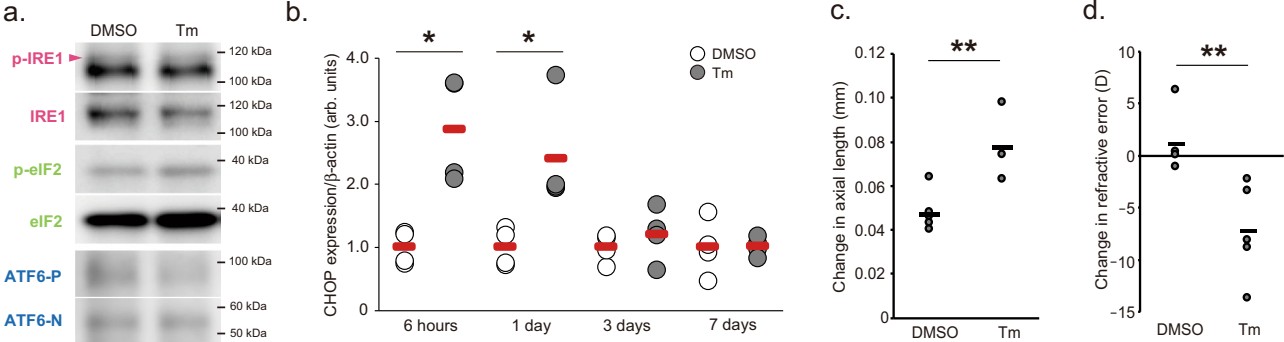

**Fig. 3 | Induction of scleral ER stress results in the development of myopia. a** A single tunicamycin (Tm; 50 µg/mL) instillation induces acute IRE1, eIF2, and ATF6 activation (1 hour after instillation) determined by Western blotting (*n* = 4, each sample pooled 3 sclerae). **b** A single Tm (50 µg/mL) instillation induces acute ER stress, evaluated as CHOP mRNA expression by qPCR (*n* = 4 per group). DMSO group in each time point were assigned a value of 1.0. *$p < 0.05$ compared to each control; Student's two-tailed *t*-test (Exact *p* values are as follows: 6 h, $p = 0.017$; 1 day, $p = 0.024$). **c** A single Tm (50 µg/mL) instillation induces axial elongation (*n* = 5 per group); **$p < 0.01$; Student's two-tailed *t*-test ($p = 0.003$). **d** A single Tm instillation induces myopic shift in refraction (*n* = 5 per group); **$p < 0.01$; Student's two-tailed *t*-test ($p = 0.009$). Source data are provided as a Source Data file.

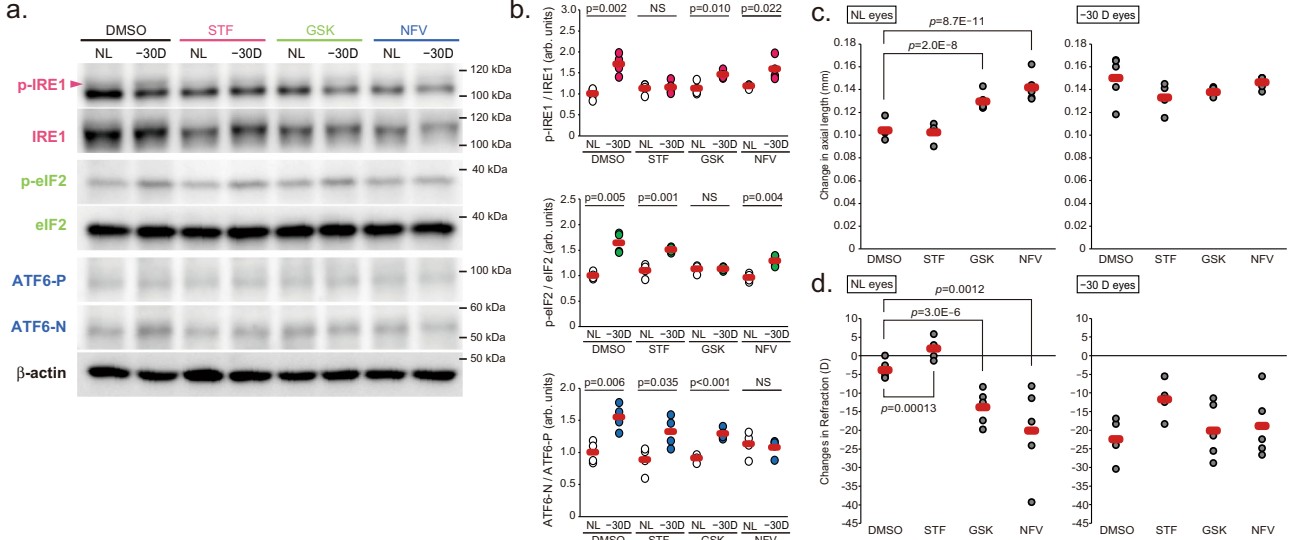

**Fig. 4 | Inhibition of a single UPR pathway with specific inhibitors disrupts the refractive condition of the mouse eyes. a** Effect of instillation of IRE1 inhibitor STF080310 (STF; 100 µM), PERK inhibitor GSK2656157 (GSK; 100 µM), and ATF6 inhibitor nelfinavir (NFV; 100 µM) on activation of the UPR pathways by LIM in sclerae. NL: No Lens control, −30D: minus 30 D lens-wearing eye. **b** Densitometric quantification of blots in Fig. 3a using ImageJ. DMSO-NL group was assigned a value of 1.0. Each group were *n* = 4 (each sample pooled 3 sclerae). The *p* values were determined by two-tailed Generalized Estimating Equations for four groups comparison (DMSO vs STF vs GSK vs NFV in each eyes) and Student's two-tailed *t*-test for no lens (NL) and −30D lens (−30D) eyes comparison in each treatments. NS: Not Significant. **c** Effects of STF, GSK, and NFV instillation on axial elongation in NL and −30 D eyes (*n* = 5 per group); ***$p < 0.001$ compared to the DMSO group; two-tailed Generalized Estimating Equations. **d** Effects of STF, GSK, and NFV instillation on refraction in NL and −30 D eyes (*n* = 5 per group); **$p < 0.01$ ($p = 0.001$), ***$p < 0.001$ compared to the DMSO group; two-tailed Generalized Estimating Equations. Source data are provided as a Source Data file.

eyes in the DMSO and STF groups (Supplemental Fig. 1a, b, d). Unexpectedly, GSK and NFV instillation induced a myopic shift in refraction, axial elongation, and attenuation of VCD + RT shortening in control, no lens-wearing eyes (Fig. 4c, d and Supplemental Fig. 6c). As a result, the myopic phenotype of intraocular differences between the control and minus lens-wearing eyes of the GSK- and NFV-treated groups were diminished (Supplemental Fig. 6a, b, d). These results suggested that myopia is induced by GSK and NFV administration, i.e., inhibition/ derangement of the PERK and ATF6 pathways. We also administered different inhibitors of each pathway (IRE1: 4µ8 C, PERK: GSK2606414, ATF6: Ceapin A-7) topically and found that the results were comparable with those shown in Fig. 4c, d and Supplemental Fig. 6a–d (Supplemental Fig. 6e–j). Based on these results, we assumed that scleral PERK and/or ATF6 dysregulation resulted in axial elongation and the myopia onset/development. Next, we examined the effect of a combination of inhibitors on the myopia development. The effects of GSK plus STF (S + G) and NFV plus STF (N + S) administration on the myopia development were comparable with the effects of instillation of GSK or NFV alone (induction of axial elongation of non-lens-wearing control eyes and loss of minus lens-dependent axial elongation) (Fig. 5c, d and Supplemental Fig. 7a–d). On the other hand, GSK plus NFV (G + N) co-instillation completely inhibited LIM-induced myopia development and showed no effects on the control eyes (Fig. 5c, d and Supplemental Fig. 7a–d). We also co-administered different inhibitors of each pathways used above and found that the results were comparable with those shown in Fig. 5c, d and Supplemental Fig. 6a–d (Supplemental Fig. 7e–j). These results were comparable with those of 4-PBA instillation (Fig. 2g, h and Supplemental Fig. 3h). We suspected that minus lens-dependent axial elongation could be suppressed by both PERK and ATF6 inhibition and axial elongation in the control eyes

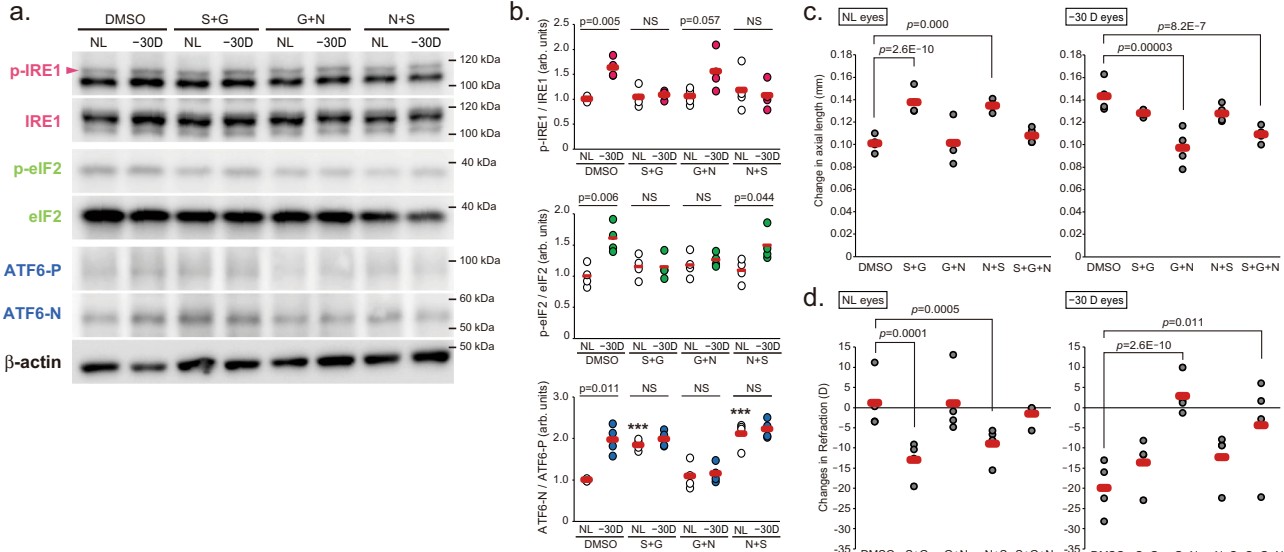

**Fig. 5 | Inhibition of both PERK and ATF6 pathways suppresses myopia progression caused by negative lens wearing. a** Effects of combined STF, GSK, and NFV instillation (S + G: STF and GSK, G + N: GSK and NFV, N + S: NFV and STF; 100 μM each) on LIM-induced activation of IRE1, eIF2, and ATF6 in sclerae. NL: No Lens control, −30D: minus 30 D lens-wearing eye. **b** Densitometric quantification of blots in Fig. 5a using ImageJ. DMSO-NL group was assigned a value of 1.0. Each group were $n = 4$ (each sample pooled 3 sclerae). The $p$ values were determined by two-tailed Generalized Estimating Equations for four groups comparison (DMSO vs S + G vs G + N vs N + S in each eye) and Student's two-tailed $t$-test for NL and −30D eyes comparison in each treatment. NS: Not Significant. **c** Effects of combined STF, GSK, and NFV instillation on axial elongation in NL and −30 D eyes ($n = 4$ per group); ***$p < 0.001$ compared to the DMSO group; two-tailed Generalized Estimating Equations. **d** Effects of combined STF, GSK, and NFV instillation on refraction in NL and −30 D eyes ($n = 4$ per group); *$p < 0.05$ ($p = 0.011$), **$p < 0.01$ ($p = 0.005$), ***$p < 0.001$ compared to the DMSO group; two-tailed Generalized Estimating Equations. Source data are provided as a Source Data file.

of the S + G and N + S-instilled groups might be due to the compensatory activation of other pathways when the inhibitors were instilled (S + G and N + S instillation resulted in ATF6 and PERK activation, respectively). Western blotting showed that enhanced phosphorylation of IRE1 and eIF2 by LIM was suppressed in the presence of their respective inhibitors (Fig. 5a, b). The ratio of ATF6-N (active form) to ATF6-P (precursor) was increased in the S + G group regardless of minus lens, indicating activation of ATF6, which may be a compensatory effect of the suppression of the IRE1 and PERK pathways (Fig. 5a, b). Surprisingly, in the N + S group, i.e., the group receiving the ATF6 inhibitor nelfinavir, activation of ATF6 was also observed regardless of lens wearing. However, axial elongation and myopia occurred in all groups in which ATF6 activation was observed (Fig. 5e−h and Supplemental Fig. 7a−d), strongly suggesting the involvement of ATF6. We also performed mixed-instillation experiments with other inhibitors (4μ8 C, GSK2606414 and Ceapin A-7), and the results were similar to Fig. 5c, d and Supplemental Fig. 7a−d (Supplemental Fig. 7e−j).

To validate the pharmacological inhibition experiments, we next performed genome editing using the CRISPR/Cas9 system to knockout the PERK and ATF6 genes in the scleral resident cells such as fibroblasts, followed by a comparison of the changes caused by LIM. Expression vector of *staphylococcus aureus* Cas9 (SaCas9) and guide RNA against *Eif2ak3* (PERK gene symbol; gEif2ak3) or *Atf6* (gAtf6) were delivered two separate adeno-associated virus (AAV, serotype: DJ) through co-injection into sub-Tenon's capsule in both eyes. First, to validate whether the serotype of AAV used in this study and the method of administration results in sufficient infection of the sclera, AAV-DJ-EGFP was injected into sub-Tenon's capsule and 28 days later, flat-mounts of the sclera were prepared and EGFP expression was assessed (Fig. 6a). EGFP fluorescence was observed in the entire sclera, especially in the posterior region, indicating that sufficient gene transfer can be expected. Next, we have observed the layered structure of the retina using SD-OCT, but the results showed no abnormalities in the retinal morphology (Fig. 6b), suggesting this procedure did not affect retina. Twenty-eight days after injection, the expression levels of

PERK and ATF6 were lower in the sclera of the mice injected with the corresponding guide RNA than in the sclera of the mice injected with scrambled, non-targeting guide RNA (Fig. 6c, d). To clarify whether knockdown of *Eif2ak3* and/or *Atf6* genes in the sclera affects the myopia onset/progression, LIM was initiated 4 days after injecting AAV-DJ-SaCas9 and each gRNA-packaged AAV-DJ. After 24 days LIM, axial elongation, refractive error and attenuation of VCD shortening were induced in the lens-wearing eyes both sham-operation group and scramble guide RNA-injected group, however, mice injected with gAtf6 or both gAtf6 and gEif2ak3 did not show those changes due to LIM (Fig. 6e−g). On the other hand, mice injected with gEif2ak3-showed myopic change with or without negative lens wearing (Fig. 6e−g), the result was concomitant with the results of PERK inhibitor instillation.

The compound AA147 (AA) and CCT020312 (CCT) were previously shown to selectively activate ATF6 and PERK in both in vivo and in vitro, respectively[21–24]. To assess whether PERK and ATF6 activation could induce axial elongation/myopia development, AA147 and CCT were topically administered to the right eyes of mice. CCT instillation enhanced phosphorylation of eIF2 and increased ATF4 expression, and AA147 instillation increased active forms of ATF6 (ATF6-N) and GRP78 expressions in the sclerae, indicating that both reagents selectively activated their targeting pathways in the sclerae (Fig. 7a−c). Then, C57BL6J mice were subjected to 100 μM CCT or AA147 instillation for 1 week. Both treatments induced a myopic shift in refraction and attenuation of VCD shortening; however, both treatments did not induce statistically significant axial elongation, albeit with a tendency (Fig. 7d, e, g, h, j, k). CCT and AA147 co-instillation strongly induced a myopic shift in refraction, attenuation of VCD shortening, and significant axial elongation (Fig. 7f, I, l). Taken together, these results suggest that among the three major effector proteins of the UPR pathway, PERK, and ATF6, possibly mainly ATF6, are involved in myopia onset/progression.

ER stress is also associated with scleral remodeling during myopia development

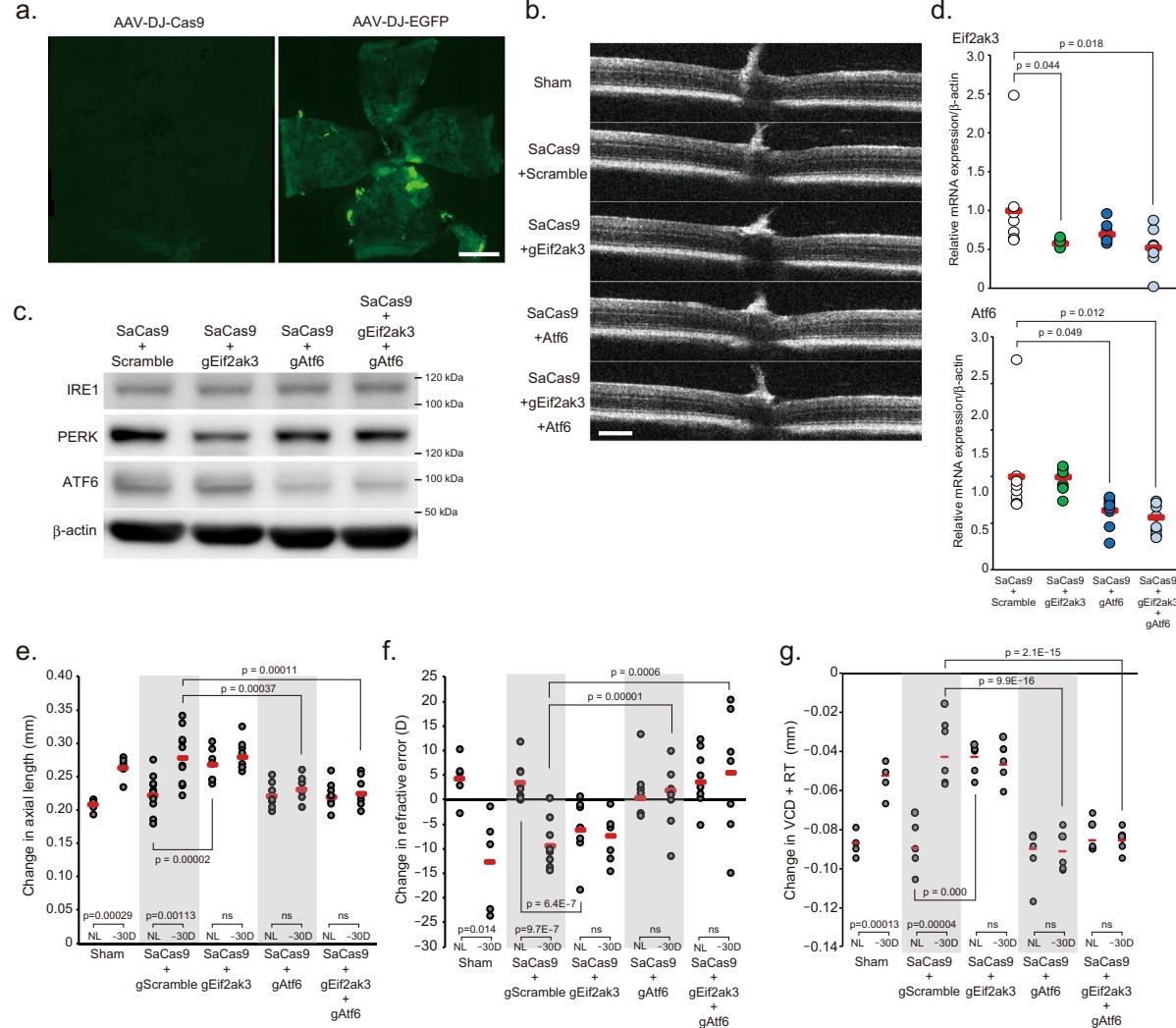

**Fig. 6 | Effect of AAV-based gene knockdown of PERK and ATF6 in sclera on myopia development. a** Distribution of EGFP expression in the scleral whole-mount 28 days after sub-tenon's capsule injection of adeno-associated virus serotype DJ (AAV-DJ)-EGFP or AAV-DJ-SaCas9 (as non-fluorescence control); scale bar, 1 mm. **b** Representative Optical Coherence Tomography images 28 days after of each AAV injection; scale bar, 100 μm. **c** Expression level of IRE1, PERK, and ATF6 protein 28 days after of AAV-SaCas9 with AAV-gRNA against *Eif2ak3* (Gene symbol of PERK, gEig2ak3) and/or *Atf6* (gAtf6) injections. **d** Scleral *Eif2ak3* and *Atf6* knockdown by AAV-DJ-CRISPR/Cas9 injection into sub-tenon' capsule. After 28 days injection, gene expression in the sclera determined by qPCR ($n = 9$, respectively). SaCas9 + gScramble group was assigned a value of 1.0. The *p* values were determined by one-way ANOVA with Tukey HSD. **e** Effects of scleral *Eif2ak3* and/or *Atf6* knockdown on axial elongation (sham: $n = 5$, gScramble: $n = 11$,

gEif2ak3: $n = 9$, gAtf6: $n = 8$, gEif2ak3 + gAtf6: $n = 8$). The *p* values were determined by two-tailed Generalized Estimating Equations for five groups comparison (sham vs gScramble vs gEif2ak3 vs gAtf6 vs gEif2ak3 + gAtf6) and Student's two-tailed *t*-test for two groups comparison (NL vs −30D). NL: No Lens control, −30D: minus 30 D lens-wearing eye. **f** Effects of scleral *Eif2ak3* and/or *Atf6* knockdown on refraction (sham: $n = 5$, gScramble: $n = 9$, gEif2ak3: $n = 9$, gAtf6: $n = 8$, gEif2ak3 + gAtf6: $n = 9$). The *p* values were determined by two-tailed Generalized Estimating Equations for five groups comparison and Student's two-tailed *t*-test for two groups comparison. **g** Effects of scleral *Eif2ak3* and/or *Atf6* knockdown on VCD + RT (sham: $n = 5$, gScramble: $n = 7$, gEif2ak3: $n = 7$, gAtf6: $n = 7$, gEif2ak3 + gAtf6: $n = 7$). The *p* values were determined by two-tailed Generalized Estimating Equations for five groups comparison and Student's two-tailed *t*-test for two groups comparison. Source data are provided as a Source Data file.

The sclera is the fibrous outer layer of the eye, mainly comprising the extracellular matrix such as collagen fibers. Thus, we assessed changes in the expression levels of collagen genes during the LIM period, because scleral remodeling is an important process for the axial elongation[25]. Among 43 collagen genes expressed in the mouse, the expression levels of *Col4a2, Col4a3, Col6a1, Col6a2, Col7a1, Col8a1, Col8a2, Col11a2, Col12a1, Col15a1,* and *Col18a1* were higher after 1-week LIM in the sclerae compared to NL sclerae (Fig. 8a). After 3 weeks of LIM, the expression level of the *Col1a1* gene was lower and that of the *Col4a3* gene was higher in the eye wearing the negative lens compared to those in the contralateral eye (Fig. 8b). To determine whether these changes in collagen gene expression depend on ER stress, we treated tunicamycin to human scleral fibroblasts (huScF) and evaluated the

changes in collagen gene expression increased by LIM for 1 week. High concentration of tunicamycin (>500 ng/mL) induced apoptosis in huScF (Supplemental Fig. 8a). Despite the ER stress occurs, the number of TUNEL-positive fibroblasts did not increase after 3 weeks of LIM in the sclera (Supplemental Fig. 2c, d). Therefore, we chose a tunicamycin concentration of 200 ng/ml. Among *Col1a1* and the abovementioned 11 LIM-responsive collagen genes, four genes (*Col4a3, Col8a2, Col11a2,* and *Col15a1*) were significantly upregulated and *Col1a1* expression was downregulated by 200 ng/ml tunicamycin treatment for 24 h (Supplemental Fig. 8b), suggesting that LIM-induced ER stress may affect the expression level of these five collagen genes. To confirm whether changes in the expression of these five genes in myopic sclerae depend on scleral ER stress, their expressions were evaluated in LIM sclerae

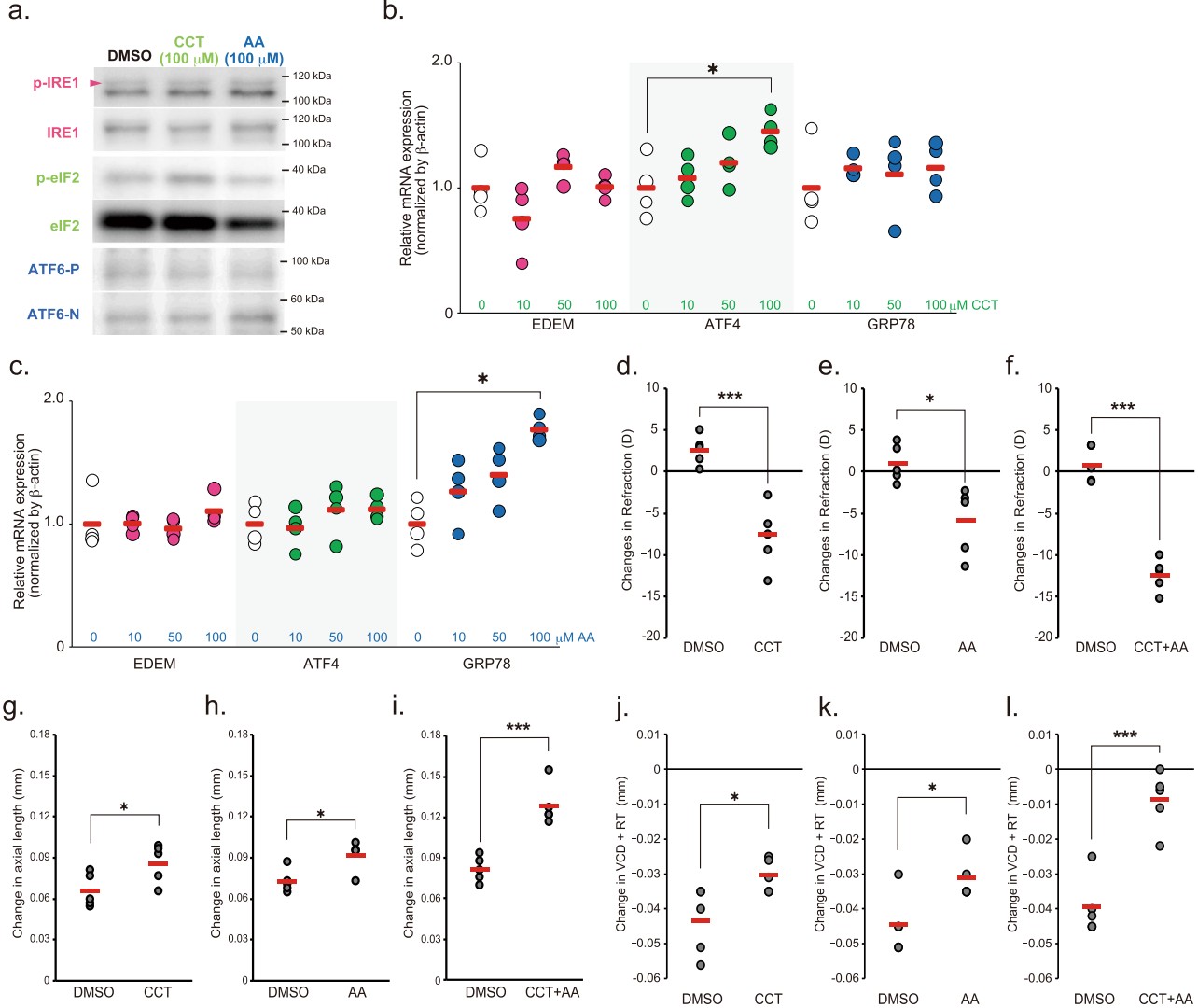

**Fig. 7 | Effect of PERK and ATF6 agonizts instillation on myopia development.** **a** Effect of PERK agonist CCT020312 (CCT) and ATF6 agonist AA147 (AA) instillation on IRE1, eIF2 and ATF6 activity in sclera. One hour after eye drop, the sclerae were harvested and subjected to Western blotting. Each compound specifically activated each pathway. **b** GRP78, ATF4, and EDEM gene expression induced by CCT. Each zero group was assigned a value of 1.0. *$p = 0.029$, compared to 0 (as a control) group; one-way ANOVA with Tukey HSD ($n = 4$, respectively). **c** GRP78, ATF4, and EDEM gene expression induced by AA. Each zero group was assigned a value of 1.0. *$p = 0.013$, compared to 0 (as a control) group; one-way ANOVA with Tukey HSD ($n = 4$, respectively). **d** Effects of 1-week instillation of CCT (100 μM) instillation on refraction; ***$p = 0.00055$; Student's two-tailed *t*-test. **e** Effects of 1-week instillation of AA (100 μM) instillation on refraction; *$p = 0.016$; Student's two-tailed *t*-test.

**f** Effects of 1-week instillation of CCT plus AA (100 μM, respectively) instillation on refraction; ***$p = 0.00001$; Student's two-tailed *t*-test. **g** Effects of 1-week instillation of CCT instillation on axial length; *$p = 0.041$; Student's two-tailed *t*-test. **h** Effects of 1-week instillation of AA instillation on axial length; *$p = 0.015$; Student's two-tailed *t*-test. **i** Effects of 1-week instillation of CCT plus AA instillation on axial length; ***$p = 0.00039$; Student's two-tailed *t*-test. **j** Effects of 1-week instillation of CCT instillation on VCD + RT. *$p = 0.036$; Student's two-tailed *t*-test. **k** Effects of 1-week instillation of AA instillation on VCD + RT. *$p = 0.024$; Student's two-tailed *t*-test. **l** Effects of 1-week instillation of CCT plus AA instillation on axial length; ***$p = 0.0004$; Student's two-tailed *t*-test. Source data are provided as a Source Data file.

treated with PBS or 4-PBA for 3 weeks and compared between groups (PBS-treated LIM eyes and fellow eyes, and 4-PBA-treated LIM eyes and fellow eyes). The expression of COL1A1 protein and *Col1a1* mRNA tended to be lower in the negative lens-wearing eye compared to NL eye in the PBS group, but there was no difference between the two eyes in the 4-PBA group (Fig. 8c, d). Furthermore, in the PBS-treated group, the expression levels of *Col4a3, Col8a2, Col11a2,* and *Col15a1* genes in LIM sclerae were higher than those in control sclerae (Fig. 8e), consistent with our experimental results (Fig. 4a). Meanwhile, the gene expression levels were comparable between LIM sclerae and control sclerae in 4-PBA-treated mice (Fig. 8e). Taken together, scleral *Col1a1, Col4a3, Col8a2, Col11a2,* and *Col15a1* expression were regulated through ER stress during the myopia progression. Collagen fiber size in the posterior region of the sclera was measured and shown in 100%

stacked bar charts (Fig. 8f, g). In the PBS-treated group, the proportion of thin fibers in LIM sclerae was higher than that in control sclerae, concomitant with a decrease in thick fibers. In contrast, the fiber proportion was comparable between LIM and control sclerae in 4-PBA-treated mice. These results indicated that the remodeling of scleral collagen during the myopia progression is mediated through ER stress, and that *Col1a1, Col4a3, Col8a2, Col11a2,* and *Col15a1* might be associated with the remodeling. Since transforming growth factor-beta (TGFβ) expression in the sclera is altered by the myopia induction[26,27] and the TGFβ-SMAD pathway regulates collagen expressions[28–31], we hypothesized that SMAD might be involved in the normalization of abnormal scleral collagen gene expression by 4-PBA. First, we examined the effect of LIM on the SMAD pathway. SMAD2 and 3 were not altered by LIM, whereas the phosphorylation level of SMAD1/5 was

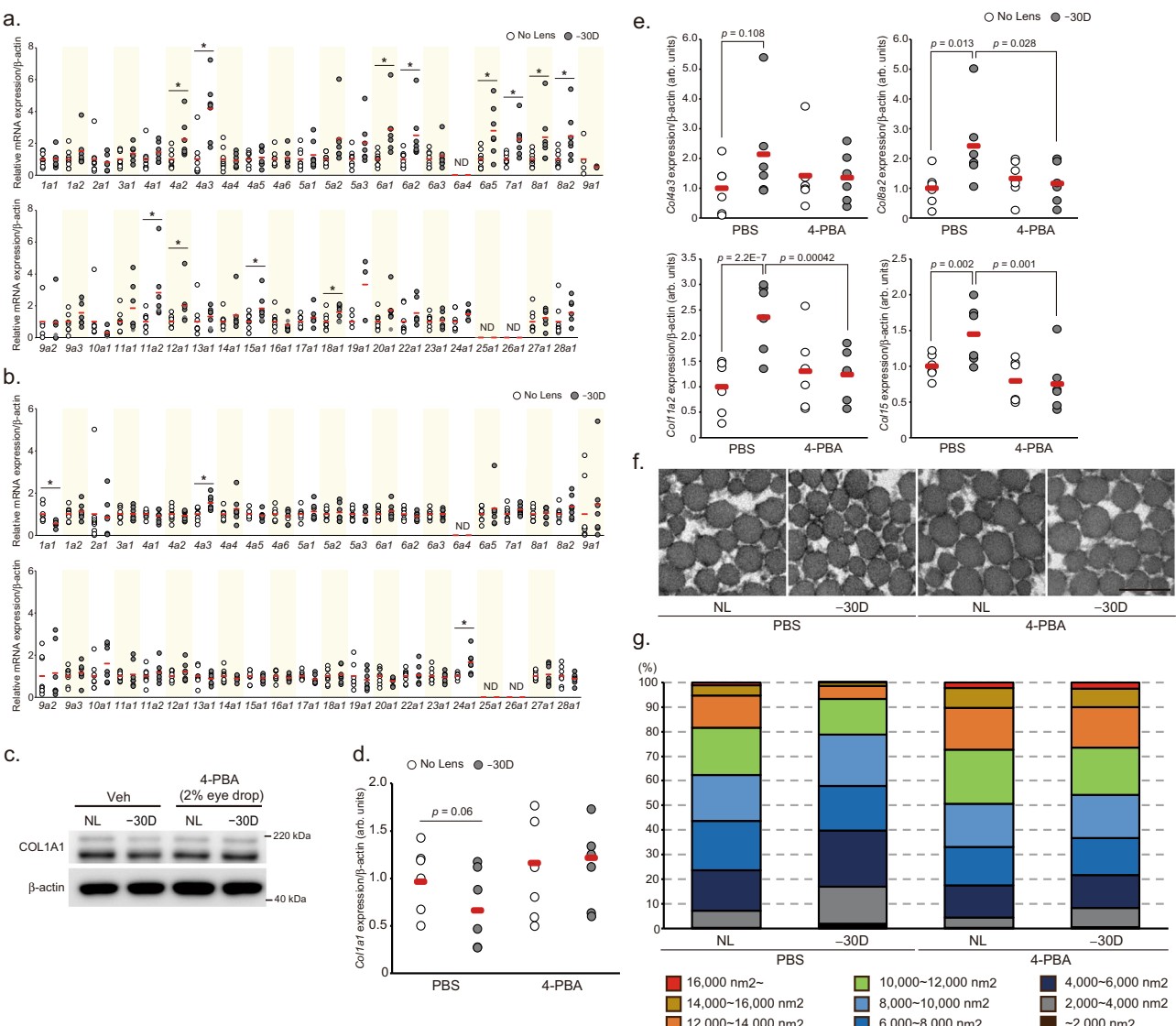

**Fig. 8 | Scleral ER stress is associated with collagen remodeling. a** Effect of 1 week LIM on the expression of 43 collagen genes in C57BL6J mouse sclerae (NL: $n = 8$, −30D: $n = 7$). No Lens group was assigned a value of 1.0. *$p < 0.05$ compared to each No Lens group; Student's two-tailed $t$-test (Exact $p$ values are as follows. 4a2: $p = 0.023$, 4a3: $p = 0.001$, 6a1: $p = 0.008$, 6a2: $p = 0.027$, 6a5: $p = 0.024$, 7a1: $p = 0.009$, 8a1: $p = 0.041$, 8a2: $p = 0.024$, 11a2: $p = 0.022$, 12a1: $p = 0.048$, 15a1: $p = 0.036$, 18a1: $p = 0.038$). **b** Effect of 3 weeks LIM on the expression of 43 collagen genes in C57BL6J mouse sclerae (NL: $n = 8$, −30D: $n = 7$). No Lens group was assigned a value of 1.0. *$p < 0.05$ compared to each No Lens group; Student's two-tailed $t$-test (Exact $p$ values are as follows. 1a1: $p = 0.012$, 4a3: $p = 0.009$, 24a1: $p = 0.017$). **c** 4-PBA instillation cancels LIM-induced decrease in COL1A1 protein expression in sclerae ($n = 4$, each sample pooled 3 sclerae). NL: No Lens control,

−30D: minus 30 D lens-wearing eye. **d** The effect of LIM and 4-PBA instillation on *Col1a1* mRNA expression in sclerae ($n = 6$, respectively). The $p$ values were determined by Generalized Estimating Equations. **e** 4-PBA instillation cancels LIM-induced increase in *Col4a3, Col8a2, Col11a2,* and *Col15a1* mRNA expression evaluated by qPCR in sclerae ($n = 6$ per group); The $p$ values were determined by two-tailed Generalized Estimating Equations. **f** Transmission electron microscopy images of sclerae collagen fiber in No lens control (NL) and −30 D lens worn (−30D) sclera of PBS or 4-PBA instilled C57BL6J mouse. Representative images of three biologically independent samples; scale bar, 200 nm. **g** 100% stacked bar charts of scleral collagen fiber area ($n = 3$ biologically independent samples). Five images from one sclera. Fiber area was measured using ImageJ software. Source data are provided as a Source Data file.

reduced by LIM (Supplemental Fig. 8c). Next, we evaluated whether 4-PBA administration counteract this change. The phosphorylation level of SMAD1/5 in the sclera of minus lens-wearing eyes was lower than that of non-wearing eyes, and it was not affected by the administration of 4-PBA (Supplemental Fig. 8d), indicating that the SMAD pathway is unlikely to be involved in myopia-induced abnormalities in scleral collagen gene expression and their correction by 4-PBA. Next, we examined the effect of pharmacological suppression of the IRE1, PERK, and ATF6 pathways on the decreased expression of COL1A1 protein in human scleral fibroblasts after tunicamycin treatment. STF had no effect on the decrease in collagen expression by tunicamycin treatment, while GSK and NFV decreased expression by their

administration itself. The administration of S + G and N + S also induced a decrease in collagen expression, but G + N completely suppressed the tunicamycin-induced decrease in COL1A1 expression (Supplemental Fig. 8e). These results are consistent with the phenotypic results of eye drop experiments with UPR inhibitors (G + N inhibit the myopia development), suggesting that PERK and ATF6 are involved in scleral remodeling in myopia progression.

## Discussion
We showed that ER stress and/or UPR pathways, especially PERK and ATF6, are also associated with an increase in the size of the eyeball during myopia development as in the other organs[7,9,11,12,14]. The strong

points of this study are that it shows these in vivo and that it shows their involvement across species.

As the individual grows, the size of the eyeball increases, and the ocular axis elongates, but this elongation is physiological and does not cause misalignment of the focus. On the other hand, axial elongation due to the wearing of negative lenses causes a misalignment of the focus plane, i.e., it creates a situation of myopia, which can be called pathological ocular axis elongation. In the present study, 4-PBA administration by both intraperitoneal injection and topical eyedrops suppressed lens-induced axial elongation but not physiological axial elongation, indicating that scleral ER stress is a fundamental regulator of only 'pathological' axial elongation. Scleral hypoxia was recently reported to be associated with ECM remodeling and the myopia development[32]. Since the hypoxic condition is known to perturb ER function, resulting in the induction of ER stress, the current findings might be associated with hypoxia response[33]. The findings together highlight scleral ER stress as a therapeutic target for pathological axial elongation in myopia. The fact that ER stress is involved only in pathological axial elongation is an extremely important finding from the viewpoint of treatment and prevention of myopia, which mainly develops and progresses in school-age children when physiological axial elongation occurs.

Administration of 4-PBA to experimental animal models of various ocular diseases has been reported to act antagonistically against these diseases, and 4-PBA is thought to have a protective effect on ocular tissues[20,34–39]. However, in the present study, we have shown that intraperitoneal administration of 4-PBA reduces retinal function analyzed by ERG measurements. The doses used in previous studies may have been relatively low (40 mg/kg/day)[37], or high doses (400 mg/kg) but single doses[36], which may have obscured the toxicity. In the present study, a dose of 200 mg/kg/day was administered for one week, which may adversely affect retinal function due to the higher dose and duration compared to the previous studies. Phenylbutyrate is rapidly converted to phenylacetic acid by beta-oxidation in vivo[40], and phenylacetic acid has long been known to have a negative effect on the brain[41,42]. Since the retina, like the brain, is a nervous tissue and large amounts of phenylacetic acid are accumulated after intraperitoneal administration of 4-PBA, large doses of 4-PBA may lead to lower visual function via accumulation of phenylacetic acid in the retina. On the other hand, ocular administration of 4-PBA showed sufficient inhibition of the myopia progression without excessive accumulation of phenylacetic acid in the retina and without deterioration of visual function, according to ERG findings. Therefore, 4-PBA eye drops appear to have very promising potential as myopia suppressant drugs.

Canonical ER stress sensor proteins, IRE1, PERK and ATF6 are activated under ER stress conditions and they independently or collaboratively regulate organ size changes associated with growth and disease. ATF6 stimulates the expression of Col II and Col X, which are differentiation markers of chondrocytes, and promotes endochondral bone growth in the longitudinal direction[14]. Schmid metaphyseal chondrodysplasia is caused by a misfolding mutation in *Col10a1*, which induces ER stress in its chondrocytes and activates IRE1, PERK, and ATF6, but the IRE1 pathway is not involved in the phenotype of short bone length[43]. Axial elongation and longitudinal bone growth are similar in terms of changes in length and in the fact that tissues consisting mainly of collagen serve as mediators. In the present study, we found that inhibition or knockdown of PERK and ATF6 by pharmacological and genetical methods induced or prevented myopia, whereas IRE1 inhibition had no effect on myopia progression. Mixed inhibitor eye drops experiments showed that simultaneous inhibition of PERK and ATF6 inhibited myopia induced by minus lens wearing, whereas other combinations did not inhibit myopia progression, but rather induced axial elongation in non-myopia induced eyes. Since the activated form of ATF6 was strongly induced in the experimental group in which axial elongation was observed regardless of the implementation

of the LIM, and knockdown of ATF6 by the CRISPR/Cas9 system suppressed axial elongation induced by the LIM, ATF6 in scleral fibroblasts appears to be a major regulator of axial elongation in myopia development. However, since axial elongation was observed in the eyes without minus lenses, whether the activity was inhibited by GSK or knocked down by the CRISPR/Cas9 system, myopia was suppressed only when both PERK and ATF6 inhibitors were applied, and both PERK and ATF6 were required to inhibit collagen degradation by tunicamycin in vitro experiments, it can be assumed that ATF6 and PERK act together to control axial elongation.

Under ER stress condition, 90-kDa form of ATF6 (ATF6-P) translocates to the Golgi, where it is cleaved by Site-2 proteinase (S2P), liberating 50-kDa N-terminal fragment (ATF6-N) can serve as a transcriptional factor in nucleus. Regarding the involvement of ATF6 in myopia, there was some discrepancy between the results using inhibitors and the results of gene knockdown using the CRISPR/Cas9 system. Ocular administration of NFV or CeapinA-7 resulted in a myopic phenotype in non-myopia-induced eyes, and the difference from myopia-induced eyes disappeared. On the other hand, ATF6 knockdown showed no myopia in either non-myopia-induced or myopia-induced eyes. The two groups differed in the phenotype of the non-myopia-induced eye but showed the same phenotype in that the left-right difference disappeared. Ceapin A-7 prevents activation of the ATF6 pathway by retaining ATF6 in the ER, i.e., by inhibiting its transport to the Golgi, thereby preventing its active form (cleaved ATF6) generation[44]. NFV, an HIV protease inhibitor, also inhibits site-2 protease, which is a key component of the cleavage (activation) of ATF6 in the Golgi apparatus[45]. Both inhibitors suppress the activity of ATF6 by preventing it from being cleaved, resulting in the accumulation of the uncleaved form (ATF6-P). On the other hand, knockdown of the ATF6 gene using the CRISPR/Cas9 system reduces its gene expression, which in turn decreases the expression level of ATF6-P, thereby attenuating the ATF6 pathway. Both interventions on ATF6 reduce the amount of ATF6-N under ER stress, but they work in opposite ways on the amount of ATF6-P. This difference in effect may have led to the difference in the results of this study, i.e., the increase in ATF6-P by drug treatment induced myopia due to its unknown function, while the decrease in ATF6-N may have suppressed the induction of myopia by negative lens wearing.

Overall, our findings highlight the importance scleral ER stress in pathological axial elongation during myopia development. The myopia-related ER stress is restricted to the sclera, is only involved in pathological axial elongation, and the chemical chaperone 4-PBA remains unmetabolized in the sclera. All of these points indicate that targeting scleral ER stress for the control of myopia progression is very promising and warrants further development.

## Methods

### Animals and minus lens-induced myopia

All animal experiments in this study were approved by the Animal Experimental Committee of Keio University (approval number: 16017) and adhered to the Institutional Guidelines on Animal Experimentation at Keio University, the ARVO Statement for the Use of Animals in Ophthalmic and Vision Research, and the Animal Research: Reporting of in vivo Experiments (ARRIVE) guidelines for the use of animals in research.

Male C57BL6J mice were housed in standard transparent cages in a temperature and humidity-controlled (24 ± 2 °C and 40–60 %) clean room under a 12 h light–dark cycle. The animals were provided with standard chow and autoclaved tap water *ad libitum*. Minus lens-induced myopia and measurement of refraction and axial length in C57BL6J mice were performed as previously reported[18,46].

Male White Leghorn chicks were purchased from Tokyo Laboratory Animals Science Co. Ltd. and housed in standard cages in a temperature-controlled clean room under a 12 h light–dark cycle.

Myopia induction and ocular parameter measurement were performed as previously reported[47].

Pharmacological ER stress suppression and induction in C57BL6J mice.

4-PBA (Cayman Chemical, MI, USA, Catalog #: 11323) and TUDCA (Sigma-Aldrich, Tokyo, Japan, Catalog #: T0266) were dissolved in PBS. STF080310 (Selleck Biotech, Tokyo, Japan, Catalog #: S7771), 4µ8 C (Selleck Biotech, Catalog #: S7272), GSK2656157 (Selleck Biotech, Catalog #: S7033), GSK2606414 (Selleck Biotech, Catalog #: S7307)), nelfinavir Mesylate Hydrate (Tokyo Chemical Industry, Tokyo, Japan, Catalog #: N0986), Ceapin-A7 (Sigma-Aldrich, Catalog #: SML2330), tunicamycin (Cayman Chemical, Catalog #: 11445), thapsigargin (Wako, Tokyo, Japan, Catalog #: 209-17281), CCT020312 (Millipore Corp, USA, Catalog #: 324879) and AA147 (Tocris Bioscience, UK, Catalog #: 6759) were dissolved in DMSO and then diluted with PBS at 1:1000. For ER stress suppression, 4-PBA (200 mg/kg body weight) or TUDCA (100 mg/kg body weight) were intraperitoneally administered daily throughout the LIM period (1 or 3 weeks). 4-PBA (0.2% or 2% solution), STF080312 (100 µM), 4µ8 C (100 µM), GSK2656157 (100 µM), GSK2606414 (100 µM), NFV (100 µM) and Ceapin-A7 (100 µM) were also instilled to both eyes throughout the LIM period. To evaluate whether 4-PBA was effective for myopia treatment, after developing myopia by LIM for 3 weeks, the minus lens and frame were removed, and then 4-PBA (2%) was instilled for 3-weeks. Axial length and refraction measurement were performed at before and after the LIM period and 1 and 3 weeks after 4-PBA instillation (Supplemental Fig. 5a).

For ER stress induction[48,49], tunicamycin (50 µg/mL) or thapsigargin (10 µM) solution was instilled in C57BL6J mice or white Leghorn chicks once. Before instillation and 1-week after instillation, axial length and refraction were measured. After a single bout of tunicamycin or thapsigargin instillation, mice were subjected to 4-PBA eye drop (2% in PBS) once a day for 1 week. CCT020312 (10, 50, and 100 µM) and AA147 (10, 50 and 100 µM) was instilled in C57BL6J mice once a day for 1-week. Before and after 1-week instillation, axial length and refraction were measured.

## Sample preparation

After experimental intervention and ocular parameter measurement, eyeballs were from C57BL6J mice and white Leghorn chicks. For transmission electron microscopic observation, mouse and chick eyeballs were fixed in ice-cold 2.5% glutaraldehyde in PBS for 1 h and cut along the sagittal plane. The cornea and lens were removed, and the rest of the tissue was further fixed in 2.5% glutaraldehyde/60 mM HEPES buffer (pH 7.4) overnight. Specimens were washed in 60 mM HEPES buffer and incubated with 1% osmium tetroxide/60 mM HEPES buffer for 2 h at 4 °C, dehydrated through an ethanol series, replaced, and embedded in Epon 812 (EM Japan, Tokyo, Japan). After embedding, the blocks were thin-sectioned at 70 nm and stained by uranyl acetate and lead citrate. Sections were visualized by JEM-1400Plus (JEOL Ltd., Tokyo, Japan).

For preparing paraffin sections (mice), eyeballs were fixed with 4% paraformaldehyde overnight, embedded in paraffin, and sliced into 5 µm sections by a microtome (REM-710, Yamato Kohki, Saitama, Japan). Sections were then stained with haematoxylin and eosin and visualized using a BX53 microscope (Olympus, Tokyo, Japan). Scleral thickness was measured using cellSens software (Olympus). The other sections were subjected to TUNEL staining and immunostaing with Bip antibody.

For mRNA expression analysis, retina and sclerae were isolated from fresh mouse and chick eyeballs and homogenized using Precellys Evolution (M&S instruments, Tokyo, Japan). Total RNA was isolated from homogenates using the RNeasy Micro Kit (Qiagen, Venlo, Netherlands). cDNA was prepared using the SuperScript IV VILO Master Mix (ThermoFisher Scientific, MA, USA) following the manufacturer's protocol. cDNA was stored at −30 °C until use.

For protein expression analysis, isolated retina and sclerae from eyeballs were homogenized in RIPA buffer (50 mM HEPES (pH 7.5), 150 mM NaCl, 1% NP-40, 0.1% sodium deoxycholate, 1 mM EDTA, 5 mM benzamidine, 10 mM β-glycerophosphate, 1 mM $Na_3VO_4$, 50 mM NaF, and 1 mM PMSF) containing Halt protease inhibitor cocktail (ThermoFisher Scientific). After centrifugation, protein concentration was measured using the BCA method and adjusted to 1.0 (retina) of 0.75 (sclera) µg/µL with Laemmli sample buffer (Nacalai Tesque). Samples were stored at −30 °C until use.

## LC-MS/MS analysis of 4-PBA

4-PBA (200 mg/kg body weight) or same volume of PBS were intraperitoneally administered for 1 week. One hour after the last dose, ocular enucleation was performed, and the retina, choroid, and sclera were isolated. Tissues from 16 animals, 32 eyes were pooled to form a single sample, and three samples were prepared for each of the PBS and 4-PBA groups. The samples were frozen with liquid nitrogen and stored at −30 °C until use.

Detection of 4-PBA in the samples using Liquid Chromatography-Tandem Mass Spectrometry (LC-MS/MS) was performed by Sekisui Medical Co., Ltd (Ibaraki, Japan). Briefly, each ocular tissue sample was homogenized by adding 9 times its weight of methanol/water for testing (1:1, v/v), respectively. The homogenates were mixed with methanol and centrifuged (100,00 × g, 4 °C, 5 min). After centrifugation, the supernatant was used for analysis. 4-Phenylbutyric Acid (Tokyo Chemical Industry Co., Ltd. Japan), Phenylacetic Acid (FUJIFILM Wako Pure Chemical Corporation. Japan) were used as standard reagents, and 4-Phenylbutyric Acid-d11 (Toronto Research Chemicals Inc. Canada) was used as internal standard reagent. To test elution patterns of 4-PBA and phenylacetic acid, standards of individual analytes and analyte mixtures at a concentration of 0.2 mg/ml in methanol. After that, 4-PBA analysis was performed on LC-MS/MS (LC-20AD systems: Shimadzu, API4000: SCIEX, TOKYO, JAPAN) with a HPLC column, Atlantis dc18 (5 µm, 2.1 mm I.D. x150 mm, Waters) and Gemini NX-C18 (3 µm, 2 mm I.D. x150 mm, Phenomenex). The mobile phase was composed by two solutions. Mobile phase A was consisted of Formic acid/water (1:10000, v/v). Mobile phase B was consisted of Acetonitrile. The gradient of each mobile phase was kept at 50% during the analysis. The injection volume was 10 µL and column temperature was 50 °C at a flow rate was 0.2 ml/min. Mass spectrometry (MS) condition was equipped with the electrospray ionization (ESI) in negative mode and the ions were monitored by Multiple Reaction Monitoring (MRM). The following optimized MS acquisition parameters were selected: collision gas, 4 psi: curtain gas, 20 psi: Ion source gas 1, 20 psi: Ion source gas 2, 20 psi. The ion spray voltage was set at −4000 V, and the source temperature was 650 °C. The following first quadrupole (Q1) to third quadrupole (Q3) transitions were monitored (m/z): $m/z$ 162.9 to 90.9 for 4-Phenylbutyric Acid, $m/z$ 134.8 to 90.8 for Phenylacetic Acid and $m/z$ 174.1 to 98.0 for 4-Phenylbutyric Acid-$d_{11}$.

## Electroretinogram (ERG)

Mice were subjected to 3 weeks application of 4-PBA by intraperitoneally or eye drop (2 % solution). Following overnight dark adaptation, scotopic and photopic ERGs were recorded as previously described[18].

AAV-based CRISPR/Cas9 system for PERK and ATF6 disruption in sclerae.

The construction of the vector carrying the *Staphylococcus aureus* Cas9 (SaCas9) or the guide RNA (against *Eif2ak3*, *Atf6* and no-targeting scramble) expression cassette and its packaging into AAV-DJ were carried out by Vector Builder (vectorbuilder.com) at final titer of AAV to >1.0 × 10^13 GC/ml. AAV packaged SaCas9 expressing vector and AAV packaged each guide RNA were mixed in a 1:1 ratio just before injection. Mice were narcotized by mixture of midazolam

(Sandoz K.K., Tokyo, Japan), medetomidine (Domitor®, Orion Corporation, Espoo, Finland,) and butorphanol tartrate (Meiji Seika Pharma Co., Ltd., Tokyo, Japan). Scopisol (Senju Pharmaceutical, Osaka, Japan) were applied to avoid reflux AAV solution. Under stereomicroscope, the conjunctiva was pinched with tweezers, the needle (33 G) was inserted, and when the tip of the needle reached the vicinity of the posterior region, the AAV solution was pushed out. The injections were given in two locations 180 degrees apart, 5 µl each, avoiding blood vessels. Eyes were covered with 0.2% polyacrylic acid every day for 4 days after injection. After that, mice were subjected to 3 weeks LIM and evaluated myopia progression, followed by harvested scleral tissues and performed qPCR analysis of *Eif2ak3* and *Atf6* to confirm whether the target gene has been knocked down.

### Scleral flat-mount
To evaluate AAV infection to sclera, 28 days after AAV-DJ-Cas9 (without EGFP) or AAV-DJ-EGFP injections, mouse ($n = 3$) were sacrificed, and the eyes were enucleated. The sclera was dissected, quartered and flat-mounted on a glass slide. The specimens were fixed by 4% PFA for 15 min at room temperature, washed three times by PBS for 5 min, and mounted with Vectashield Mounting Medium with DAPI (Vector Laboratories, Burlingame, CA, USA) and glass coverslips. EGFP signal was visualized with a BZ-9000 system (Keyence, Tokyo, Japan)

### Western blotting
Protein (7.5 of 10 µg/well) was resolved by SDS-PAGE, transferred to PVDF membranes (Merck Millipore, MA, USA), blocked with Blocking One (Nacalai Tesque, Tokyo, Japan), and incubated overnight with anti-ATF6 (1:1000, 73-505, Bio Academina, Osaka, Japan), IRE1 alpha (phosphor Ser724) antibody (1:1000, GTX132808, GeneTex, CA, USA), IRE1 (14C10) Rabbit mAb (1:1000, #3294), phosphor-eIF2α (Ser51) (1:1000, D9G8) XP Rabbit mAb (1:1000, #3398), eIF2α (D7D3) XP Rabbit mAb (1:1000, #5324), phospho-Smad1/5 (1:1000, Ser463/465) (41D10) rabbit mAb (1:1000, #9516), Smad1 (D59D7) XP Rabbit mAb (1:1000, #6944), phosphor-Smad2 (S465/467)/Smad3 (S423/425) (D27F4) Rabbit mAb (1:1000, #8828), Smad2/3 (D7G7) XP Rabbit mAb (1:1000, #8685) and β-actin (8H10D10) Mouse mAb (1:5000, #3700) (Cell Signaling Technologies Japan, Tokyo, Japan) antibodies at 4 °C. Membranes were incubated with appropriate HRP-conjugated secondary antibodies (1:10000) and visualized using SuperSignal West Femto Maximum Substrate (ThermoFisher Scientific). SDS-PAGE was performed in 10% acrylamide gels with protein size marker (Magic-Mark XP Western Protein Standard, ThermoFisher Scientific). Densitometric analysis was performed using ImageJ 1.53 software.

### Quantitative PCR
Quantitative real-time PCR was performed with *PowerUp* SYBR Green Master Mix (Applied Biosystems, CA, USA) using a StepOnePlus real-time PCR system. Expression levels were normalized to that of β-actin. Primer sequences are shown in Supplemental Tables 1–3.

### Immunohistochemistry and TUNEL assay
Paraffine sections were fixed with 4% paraformaldehyde for 10 min at room temperature. After three washes with PBS, the sections were incubated in 0.1% Triton-X100 in PBS for 15 min, blocked by 10% goat serum for 1 h, and incubated with an anti-mouse BiP (C50B12) Rabbit mAb (1:100, #3177, Cell Signaling Technologies Japan). Then, the sections were washed with PBS-T and incubated with an Alexa Fluor 555-conjugated anti-rabbit IgG (1:200, Life Technology Japan, Tokyo, Japan) for 30 min at room temperature. Sections were washed with PBS-T and mounted with Vectashield Mounting Medium with DAPI (Vector Laboratories, Burlingame, CA, USA) and glass coverslips. Images were captured with a BZ-9000 system (Keyence, Tokyo, Japan) and quantification analysis were performed using hybrid cell count application ($n = 5$ in NL group, $n = 6$ in −30 D group).

TUNEL assay was performed using an in situ Apoptosis Detection Kit as per the manufacturer's protocol (TAKARA Bio, Shiga, Japan). Three 20x magnified images near the posterior region were obtained from each section (4 sections in total), and the number of TUNEL-positive and DAPI-positive nuclei was measured manually.

### Cell culture
Primary human scleral fibroblasts (huScF) were purchased from Lifeline Cell Technology (Frederick, MD, USA). The huScF were grown in FibroLife S2 Fibroblast Medium (Lifeline Cell Technology). Cells were subjected to tunicamycin treatment (10 ng/mL to 10000 ng/mL) for 24 h, after which cell death was evaluated on TUNEL assay using In situ Apoptosis Detection Kit (Takara, Kyoto, Japan) as manufactures protocol. For the analysis of collagen gene expression by Tm, after treatment with 200 ng/ml of Tm for 24 h, RNA was collected from the cells as described above, cDNA was synthesized, and qPCR was performed using primers listed in Supplemental Table 3. Expression levels of ER stress marker genes and collagen genes were measured by qPCR as described above.

### Statistical analysis and reproducibility
Sample size calculation was performed by G*Power software ver 3.1.9. SPSS statistics 27 software was used to analyze all data.. Differences between groups were analysed using Student's *t*-test or one-way analysis of variance with Tukey HSD or Generalized Estimating Equations. In the GEE analysis, the multiplicity was adjusted using Sidak's method. A *p*-value of less than 0.05 was considered to denote a statistically significant difference. All exact *p*-values are given in Figures or in Figure legend, and values are given in decimal notation to the fifth decimal place and in exponential notation to the sixth decimal place and below (i.e., 0.0000006 is expressed as 6.0E-7). Values followed by 0 to the 20th decimal place are shown as $p = 0.000$. All experiments including TEM, IHC, TUNEL staining, SD-OCT and Western blotting, were representative of at least two independent repeats to confirm reproducibility.

### Reporting summary
Further information on research design is available in the Nature Research Reporting Summary linked to this article.

## Data availability
All data are available within the manuscript as Supplementary Information or Source Data files. Source data are provided with this manuscript.

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

## Acknowledgements

We thank K Mori, T Ishikawa, and T Okada (Department of Biophysics, Graduate School of Science, Kyoto University) for meaningful discussions and suggestions, and T Nagai (Laboratory of Electron Microscopy, Keio University School of Medicine) for support with TEM observation, and Y Sato and K Nagashima (Clinical and Translational Research Center, Keio University Hospital) for support with Statistical analysis, and K Kurosaki and A Kawabata for their excellent technical assistance. We are also grateful to the Collaborative Research Resources, School of Medicine, Keio University, for technical support and reagents. This work was supported by grants (S. I.) from KAKENHI (Grant-in-Aid for Scientific Research (B): 16H03258 and Grant-in-Aid for Scientific Research (C): 20K09834) from the Ministry of Education, Culture, Sports, Science and Technology of Japan and AMO Japan's Contracted Research Grant (AS2020A000130617). This work was also supported by a grant for myopia research from Tsubota Laboratory, Inc. (Tokyo Japan).

## Author contributions

K.T. and T.K. conceived the concept of the study and supervised the study. S.I. carried out all experiments and wrote the manuscript. K.N. made a critical revision of the manuscript. X.J. and H.T. developed and prepared LIM mice and LIM chicks. Y.M. measured mouse ERG and prepared Supplemental Fig. 4a–h. D.L. performed the analysis of retinal samples. N.S. conducted LC-MS/MS experiments and prepared Supplemental Figs. 9–10 and Supplemental Tables 4–7. H.J. performed in vitro study using scleral fibroblasts. K.M., Y.K., H.K., N.O., M.I., and C.S. measured axial length and refraction.

## Competing interests

We report grants from Tsubota Laboratory and Novartis during the course of the study. In addition, K.T., T.K., S.I., and X.J. have been internationally applying for a patent WO2018/164113, which has already been registered in Japan. The sponsors had no control over the experiments, interpretation, writing, or publication of this work. K.T. reports serving as chief executive officer for Tsubota Laboratory, a company producing myopia-related devices. H.T., T.K., and K.T. own unlisted stocks of Tsubota Laboratory. The other authors declare no competing interests.
