## [Peer Review File · Nature Communications]

Scleral PERK and ATF6 as targets of myopic axial elongation of mouse eyesReviewers' comments:

Reviewer #1 (Remarks to the Author):

This manuscript describes a number of pharmacological experiments that support an association between scleral ER stress expression and development of myopia in a murine myopia model. The authors demonstrated two types of ER transmembrane proteins, ATF6 and PERK, were activated in sclera in myopia development. They also showed that activation of scleral ER stress caused myopia development. In contrast, PERK and ATF6 inhibition reduced pathological myopia progression. A chick myopia model was also used in some experimental sets to confirm the generality of their conclusions.. An in vitro model using a fibroblasts cell line was also used to study if the collagen expression was depending on ER stress. The authors concluded that ER stress could be a promising therapeutic target for inhibiting maladaptive increases in optical axis elongation and pathological myopia progression.

A similar study using a Guinea Pig model have revealed the potential involvement of ER stress in scleral remodeling in myopia progression (J Ophthalmol. 2020 Jun 10;2020:3264525.). Although the experimental data is very impressive in its scope and execution, there are several fundamental flaws with the current study and the conclusions reached need to be better argued.

Major Points:

- 1)Based on the experimental design used in similar studies to delineate mechanisms underlying myopia progression, age-matched control mice are needed to draw a definitive conclusion. For instance in Figure 1c-e. in order to check for a yoking effect on LIM in mice, the scleral expression levels of ER stress markers are evaluated in age-matched control mice. This procedure provides another relevant control group.
- 2)It is difficult to see differences in the scleral expression levels of ATF6, pIRE1/IRE1, and P-eif2/eif2 between NL and -30D groups in Figure 1d. In addition to p-eif2a, the PERK expression level should also be included. The authors need to provide more representative blots, and they should give quantitative densitometric measurements used to draw their conclusions.
- 3)At present, the statistical reporting is still inadequate even though authors have performed many statistical tests in analyzing their data. The statistical methods used to analyze the data should be mentioned in the corresponding legends. In some cases, it doesn't appear that the correct statistical test was used. For example, a two-way ANOVA should be used to analyze the data in Figure 2a, 3a-d.
- 4)It is relevant to also determine if stress-induced changes in the ER- also occurred in the retinal layer in myopia-induced eyes. Most of the pharmacological experiments in the current study were performed by intraperitoneal injection, which may also affect other ocular tissues. Furthermore, the sclera is avascular in the guinea pigs, which could delay drug buildup in the sclera relative to that in vascularized tissues such as the retina, and cornea. Therefore, the authors need to provide data to prove if these signaling pathways were actually activated or inhibited by the agonists and antagonists. For example, the authors need to assess if PERK, ATF6, and IRE1 were actually inhibited in the sclera after the drug treatment shown in Figure 3.
- 5)3T3: The authors need to explain why this irrelevant cell line was used. Primary scleral fibroblasts are a better choice.

Minor points:

- 1) IHC: The authors' conclusions about the alterations in the expression of BIP and TUNEL are based on changes in the immunohistochemical staining. The IHC quantification of fluorescence intensity measurements over the same ROI size in each image, needs to be provided in the Extended Data Figure 2. Additionally, the authors have not provided any controls for the IHC, such as no-primary controls to evaluate background staining. Please label the Figures to show the orientation of the tissue (is the bottom part facing towards the choroid?). Does the Figure show the whole scleral region that was analyzed or only a specific region?
- 2) RT-PCR: The authors need to explain why two different reference genes, GAPDH (Figure 1e) and β -actin (Figure 3e-g) were used for normalization purposes in similar experiments. In addition, a previous study showed the significance of scleral hypoxia in ECM remodeling during myopia development (Proc Natl Acad Sci U S A. 2018 Jul 24;115(30):E7091-E7100) and the authors speculated that hypoxia was a potential trigger of ER stress. Under the authors' hypoxic condition, it was not the best idea to use GAPDH as the reference gene. In addition, the reference values used to normalize values in figures showing relative expression level data should be given in the legends.
- 3) Western blot: Quantitative densitometric data should be included in the Figures throughout the manuscript. And molecular mass markers should be shown in the western blots.
- 4) In myopic eyes, the axial elongation was mainly due to an increase in vitreous chamber depth. Please provide the vitreous chamber depth data throughout the manuscript.

Clarifications needed:

- 1) Figures 1. Panels f and g are only described in the legend, but the actual data is not shown. .
- 2) Figures 1c. It is difficult to see the rough endoplasmic reticulum (RER) in the control eyes. Perhaps a higher magnification image is needed (refer to J Ophthalmol. 2020 Jun 10;2020: PMID 3264525.)?.
- 3) The figures do not follow the same sequence as presented in the text. For example, Figure 2f-h (line 91) appeared before Figure 2d-e (line 104). Other sections are similarly disordered and confusing.

Reviewer #2 (Remarks to the Author):

The authors found that ER stress might involve in LIM. In addition, considering that intraperitoneal injection and eye drop of 4-PBA (ER stress suppressor) inhibited LIM-pathologies, this study is very attractive because there are no therapeutic agents for myopia. The author also identified PERK and ATF6 as the therapeutic target. However, this study has the following issues:

[Major issues]

1. This reviewer cannot recognize the novelty of this study because Zhu C. et al. have already reported that ER stress involves in scleral remodeling in a Guinea pig model of form-deprivation myopia and that 4-PBA has the potential to decrease COL1A1 in Guinea pig scleral fibroblasts treated tunicamycin. [Zhu C, Chen Q, Yuan Y, Li M, Ke B. Endoplasmic Reticulum Stress Regulates Scleral Remodeling in a Guinea Pig Model of Form-Deprivation

Myopia. *J Ophthalmol.* 2020;2020:3264525. Published 2020 Jun 10.

doi:10.1155/2020/3264525.] This reviewer asks the authors to revise the abstract to convey the novelty.

2. In this study, the authors discovered the involvement of ER stress in myopia. However, the reason why the authors focused on ER stress is lack. This reviewer hopes to add the description. For example, are there any reports that ER stress (such as ATF6 or PERK activation) would be observed in the sclera of pathological myopia patients?

3. In Figure 1c, this reviewer recognized the dilated endoplasmic reticulum (ER) in LIM eye. However, this reviewer did not understand the ER in the control eye. To compare the normal ER with the pathological one, the authors had better show the ER in the control eye using the mark such as asterisk.

4. In pages 5, lines 63-65 (Figure 1d); This reviewer cannot admit the description because this reviewer cannot see the increasing of ER-stress related proteins in the LIM group based on only their typical images. Considering the importance of Figure 1d in this study, this reviewer asks the authors to replace the typical images to support the authors' conclusion and to quantify the immunoblots of Figure 1d. In addition, the immunoblots of cleaved ATF6 are pale color. Please improve the typical images.

5. In Figure 2, the authors showed the potential of 4-PBA to inhibit LIM development by intraperitoneal injection and eye drop. However, this reviewer doubts the following issues: first, were 4-PBA actually delivered to sclera by intraperitoneal injection in Figure 2b? Did 4-PBA intraperitoneal administration show some adverse effects? Second, how about the influence of 4-PBA eye drop on anterior segment such as the thinning of lens? Third, does 4-PBA decrease the proteins related PERK and ATF6 pathway? The authors should evaluate whether 4-PBA might decrease the expression of cleaved ATF6 and PERK activation.

6. In Figure 1e and 2a, what is the method of statistical analysis? Is it one-sided test or two-sided test? This reviewer doubts that some ER stress markers have significant differences considering the statistical variability. In addition, the asterisks, which means significantly differences, should be marked on the column of 4-PBA in Figure 2a.

7. The authors demonstrated that GSK2656157 (PERK inhibitor) plus nefinavir (ATF6 inhibitor) co-instillation completely inhibited LIM development (Figure 3c and 3d). On the other hand, the authors did not evaluate whether GSK2606414 (PERK inhibitor) plus Ceapin-7 (ATF6 inhibitor) would inhibit LIM development in Extended Data Figure 6. This reviewer asks the authors to evaluate whether GSK2606414 plus Ceapin-7 also have the potential to inhibit LIM development. This data might be important to confirm that PERK/ATF6 pathway involves in LIM development.

8. The author demonstrated that ATF6 and PERK would involve in LIM pathology by pharmacological approach. However, considering of many compounds has some unexpected effects, this reviewer suggests the authors to demonstrate them by another approach. For example, using shRNA or siRNA of ATF6 and PERK might be important to confirm that ATF6 and PERK might involve in LIM pathology.

9. In Extended Figure 2e (and 2f), the authors described that cell death was not occurred in LIM. However, the authors supplied only the typical images. Therefore, this reviewer asks the authors to quantify the TUNEL-positive cells.

10. The involvement of ER stress on collagen expression is unclear in this study. TGF β -SMAD signaling pathway is one of the major pathways to express collagen. Therefore, this reviewer asks the authors to evaluate SMAD activation on the sclera of LIM group and 4-PBA treated group.

11. In this LIM model, posterior staphyloma, which causes pathologic myopia, is observed? If so, is posterior staphyloma decreased by 4-PBA treatment? Considering of formation of posterior staphyloma is a key component of a spectrum of vision-threatening myopic maculopathies, this reviewer is interested in it.

[Minor issues]

1. In the legend of Figure 1, the authors described Figure 1f and 1g. However, this reviewer could not find Figure 1f and 1g. The authors should address this issue.

2. Please spell out "NL" in Figure 1 legends.

3. Please magnify the font size of Figure 1e.

4. The authors should revise the following misprints:

In pages 6, lines 91; "Fig. 2f" \rightarrow "Fig. 2d"

In pages 6, lines 95; "Fig. 2g and h" \rightarrow "Fig. 2e and f"

In pages 7, lines 103-104; "Fig. 2d and e" \rightarrow "Fig. 2g and h"

5. In Figure 2g and 2h, what for animals were 0.2% and 2% 4-PBA applied? LIM treated group or Non-LIM treated group? Please add the description.

6. In this study, the authors used many kinds of compounds to regulate ER-stress. Therefore, this reviewer recommends the authors to make the table showing their target to enable to read easily such as STF083010: IRE1 inhibitor.

7. In pages 11, lines 174; Extended Data Fig. 2f is lack in this manuscript.

8. In Extended Data Figure 5b: the vertical line meaning Δ Axial length is hidden.

9. In Extended Data Figure 5h: Please modify "p=0114".

Reviewer #3 (Remarks to the Author):

Review overview: Dr. Shin-ichi and colleagues identified that PERK and ATF6 control axial elongation of myopia pathology in LIM mice model. They build on interesting model described in previous report (Jiang X., 2018), as well as pharmacological treatments of ER stress inhibitor and activator modulate scleral physiology of LIM. Interestingly, their EM study also revealed enlarged rough ER in scleral fibroblast of LIM mice. Dr. Shin-ichi et al also tested that ER stress reducing compound, 4-PBA, attenuated LIM-derived ER stress associated gene expression and ER stress inducing compounds, Tg and Tm, exacerbate

pathological myopic phenotypes. While this work might translate the role of ER stress as a key player in myopia progress, several concerning points should be revised, including lack of proper control and statistical analysis, rough description of figure legend, irrational statistical interpretation, repeated wrong numberings of figure in manuscript and overinterpretation of the mechanism related to ER stress:

1. In Fig 1c, they showed the enlarged rough ER in sclera of LIM eye. Is it shown only in sclera or, in the other region of eyeball layer? Please add the result of whether LIM induced-enlarged ER is only shown in sclera or the other layer. The EM images are very difficult to interpret subcellular organelles because they appear out-of-focus or are too low-magnification.
2. Please add detail information of antibody catalog number in method section.
3. In Fig 1d, it is hard to see the difference of ER stress protein level, phospho-IRE1, not like their manuscript description. Please add the statistical analysis graph by normalizing with suitable control protein. In addition, phospho-eIF2a seems to be decreased by normalization of eIF2a. Please discuss about the point. Last, weight markers are needed to confirm bands are correct size.
4. In Fig 1e, please add information of what statistical analysis they used is to show the significance and how the significance is calculated by comparison of what.
5. In Fig 2a, they showed 4-PBA suppressing UPR gene expression. However, it is not clear whether it is for 1 or 3 weeks. Please add the detail information of experiment time frame.
6. Fig 2 a,b,g,h seem to be replication with Extended Data Fig 4 a,b,g,h. Please replace the Fig 2 a,b,g,h by Extended Data Fig 4 a,b,g,h. Otherwise, please address why you displayed like that.
7. In Fig 2d, they need to mention what kind of chop expression it is. Is it RNA or protein? Also, they need to describe what LIM time period of sample they tested.
8. Please correct figure numbering (Fig.2f, 2g, 2h, Extended Data 4b~f, 4g~k, and so on). There is no Extended Data 4k on figure, not like manuscript description.
9. In Fig 3 a~d, I am doubt if their data interpretation of statistic comparisons is reasonable. They should compare the result of each compound in that of same model, or each chemical in same mouse, not like comparing with only NL/DMSO. To conclude the effect of ER stress inhibitors, they should redo statistic analysis comparing with an eligible control.
10. They need to add primer information for qPCR. They missed the Extended Table 1 for the information, not like their description on method section.
11. In Fig 3e, f, and g, they should mention concentration information of each chemical.
12. In Fig 3e~m, they should mention what model they compared between, NL or -30D.
13. In Fig 4, they use the word, moderate ER stress, in their Tm-NIH 3T3 cell culture model. But, they did not show how they determined the moderate concentration of ER stress in mice model. Using Tm dose-dependent curve test, please add the result of Tm-induced ER stress in LIM model

Response to Reviewer #1

We wish to express our strong appreciation to you for your insightful comments on our paper. We feel the comments have helped us significantly improve the paper. Based on your comments, we have performed additional experiments and additional analysis as follows:

Major Comment 1:

Based on the experimental design used in similar studies to delineate mechanisms underlying myopia progression, age-matched control mice are needed to draw a definitive conclusion. For instance, in Figure 1c-e. in order to check for a yoking effect on LIM in mice, the scleral expression levels of ER stress markers are evaluated in age-matched control mice. This procedure provides another relevant control group.

Response: We appreciate your comment on this point. We searched for previous published papers and found papers that provide age-matched controls as you mentioned. Thanks to this comment, we have learned something new. We conducted an additional series of experiments with age-matched controls, and Fig. 1d-f and Supplemental Fig. 2e-g2 have been changed as shown below.

Fig. 1: Effect of LIM on UPR pathways and their target genes expression in the sclera (d-f). We redid LIM experiments with age-matched control and performed quantitative densitometric measurements.

Major Comment 2:

It is difficult to see differences in the scleral expression levels of ATF6, pIRE1/IRE1, and P-eif2/eif2 between NL and -30D groups in Figure 1d. In addition to p-eif2a, the PERK expression level should also be included. The authors need to provide more representative blots, and they should give quantitative densitometric measurements used to draw their conclusions.

Response: In accordance with this comment 2 and above comment 1, we performed additional experiments with age-matched controls, and representative blots are shown in Fig. 1d, and results of quantitative densitometric measurements using image J are shown in Fig. 1e. In addition, quantitative analysis of the results of Western blotting has been added throughout this paper as shown in response to Major comments 1.

Major Comment 3:

At present, the statistical reporting is still inadequate even though authors have performed many statistical tests in analyzing their data. The statistical methods used to analyze the data should be mentioned in the corresponding legends. In some cases, it doesn't appear that the correct statistical test was used. For example, a two-way ANOVA should be used to analyze the data in Figure 2a, 3a-d.

Response: In accordance with this comment, the statistical methods used in the analysis were described for all legends, and the statistical processing methods were reanalyzed using Generalized Estimating Equations (a method of analysis similar to two-way ANOVA) for the parts that were pointed out.

Major Comment 4:

It is relevant to also determine if stress-induced changes in the ER- also occurred in the retinal layer in myopia-induced eyes. Most of the pharmacological experiments in the current study were performed by intraperitoneal injection, which may also affect other ocular tissues. Furthermore, the sclera is avascular in the guinea pigs, which could delay drug buildup in the sclera relative to that in vascularized tissues such as the retina, and cornea. Therefore, the authors need to provide data to prove if these signaling pathways were actually activated or inhibited by the agonists and antagonists. For example, the authors need to assess if PERK, ATF6, and IRE1 were actually inhibited in the sclera after the drug treatment shown in Figure 3.

Response: In accordance with these comments, several additional experiments were performed, and the following data and interpretations have been added to our manuscript as shown below.

“On the other hand, in the retinal layer, there was no LIM-induced activation of IRE1, eIF2, ATF6, or increased expression of their downstream genes (Supplemental Fig 2e and f). ER of the retinal pigment epithelium (RPE) was observed by TEM, but no expansion of the ER was observed (Supplemental Fig 2g), suggesting that LIM does not cause ER stress in the retinal layer.”

Supplemental Fig. 2:
Effect of LIM on UPR pathways and their target genes expression in the retina(e-g).
ER stress did not occur in the retinal layer

The absence of LIM-induced ER stress in the retinal layer was confirmed by Western blotting of IRE1, eIF2, and ATF6 (Supplemental Fig. 2e), qPCR of their target gene expression levels (Supplemental Fig. 2f), and TEM of RPE endoplasmic reticulum morphology (Supplemental Fig. 2g; Since our TEM specimens were limited to the RPE and the outer segment of rod cells in the retinal layer, we observed the RPE).

Regarding the effects of i.p. administration of 4-PBA on other ocular tissues, ERG was measured to verify the effect on retinal function, and it was shown that i.p. administration may have a negative effect on retinal function (Supplemental Fig. 4a-d, shown in the right figures), so this comment makes it clear that ocular administration is preferable for clinical application because 4-PBA instillation had no harmful effect to ERG and were sufficiently effective in reducing myopia progression.

Furthermore, as a result of LC-MS/MS analysis of ocular tissues (retina, choroid, and sclera) to determine whether the administered 4-PBA properly reached the sclera, the amount of 4-PBA was highest in the sclera when administered intraperitoneally (Supplemental Table 1). This is because, as you pointed out, vascularized tissues (retina and choroid) receive more 4-PBA, but both are metabolically active tissues, so 4-PBA undergoes β -oxidation and is immediately converted to phenylacetic acid (Supplemental Table 2

showed the presence of large amounts of phenylacetic acid in both tissues), while the buildup of 4-PBA in sclera more slowly but remains 4-PBA for a longer time due to its relatively low metabolism, resulting in a situation that is highly favorable for targeting the reduction of scleral ER stress. Consistent with these results, intraperitoneal administration of 4-PBA inhibited LIM-induced activation of IRE1, eIF2, and ATF6 (Fig. 2a and b), and ocular administration of 4-PBA or agonists induced inhibition/activation of their respective target

Supplemental Fig. 4a-d: Effect of i.p. 4-PBA injection on retinal function.

Intraperitoneal 4-PBA injection showed harmful effect on the retina determined by ERG measurement.

Table 1 Concentrations of 4-Phenylbutyric Acid in mouse ocular tissues after 4-Phenylbutyric Acid treatment

Treatment	Sample	Concentration of 4-Phenylbutyric Acid (ng/g)				
		1	2	3	Mean	SD
4-Phenylbutyric Acid	Retina	209	221	165	198	29
	Choroid	327	520	208	352	157
	Sclera	829	691	117	546	378
PBS	Retina	BLQ	BLQ	BLQ	BLQ	NC
	Choroid	BLQ	BLQ	BLQ	BLQ	NC
	Sclera	BLQ	BLQ	BLQ	BLQ	NC

BLQ : Below the lower limit of quantification

Table 2 Concentrations of Phenylacetic Acid in mouse ocular tissues after 4-Phenylbutyric Acid treatment

Treatment	Sample	Concentration of Phenylacetic Acid (ng/g)				
		1	2	3	Mean	SD
4-Phenylbutyric Acid	Retina	34100	33900	30900	33000	1800
	Choroid	7780	5120	6040	6310	1350
	Sclera	1610	1480	1500	1530	70
PBS	Retina	BLQ	BLQ	BLQ	BLQ	NC
	Choroid	BLQ	BLQ	BLQ	BLQ	NC
	Sclera	BLQ	BLQ	BLQ	BLQ	NC

BLQ : Below the lower limit of quantification

NC: Not calculated

molecules (Fig. 2e, f, I; Fig. 3a, k; Supplemental Fig. 6a).

Fig. 2: Effect of 4-PBA or tunicamycin administration on activation of IRE1, PERK and ATF6 pathways.

a and b: results of i. p. injection

e and f: results of instillation

i: effect of Tm instillation

4-PBA suppressed LIM-induced activation UPR pathways.

Major Comment 5:

3T3: The authors need to explain why this irrelevant cell line was used. Primary scleral fibroblasts are a better choice.

Response: In accordance with this comment, we bought primary human scleral fibroblasts and all the *in vitro* experiments were redone using them (Supplemental Fig. 7a, b, d).

Minor Comment 1:

IHC: The authors' conclusions about the alterations in the expression of BIP and TUNEL are based on changes in the immunohistochemical staining. The IHC quantification of fluorescence intensity measurements over the same ROI size in each image, needs to be provided in the Extended Data Figure 2. Additionally, the authors have not provided any controls for the IHC, such as no-primary controls to evaluate background staining. Please label the Figures to show the orientation of the tissue (is the bottom part facing towards the choroid?). Does the Figure show the whole scleral region that was analyzed or only a specific region?

Response: In accordance with this comment, we re-performed IHC and TUNEL staining and quantified fluorescence intensity measurements in BiP staining and TUNEL assay (Supplemental Fig. 2a-d). And, we added no-primary controls in IHC experiment (Supplemental Fig. 2a) and labels to show the orientation of the tissue in figure legends as shown below.

“The area between the yellow dot lines is the sclera (S), and the area between the yellow and green dot lines is the choroid (C) (legend of Supplemental Fig. 2a)”

“The bottom part facing towards the choroid (Legend of Supplemental Fig 2c)”

Supplemental Fig. 2: **Effect of LIM on Bip expression (a,b) and number of TUNEL-positive cells (c,d)**

We performed quantification of fluorescence and added the results in Supplemental Fig. 2b and d

Minor Comment 2:

RT-PCR: The authors need to explain why two different reference genes, GAPDH (Figure 1e) and β -actin (Figure 3e-g) were used for normalization purposes in similar experiments. In addition, a previous study showed the significance of scleral hypoxia in ECM remodeling during myopia development (Proc Natl Acad Sci U S A. 2018 Jul 24;115(30):E7091-E7100) and the authors speculated that hypoxia was a potential trigger of ER stress. Under the authors' hypoxic condition, it was not the best idea to use GAPDH as the reference gene. In addition, the reference values used to normalize values in figures showing relative expression level data should be given in the legends.

Response: In accordance with this comment, the internal standard was changed to Hprt (qPCR for the retina, Supplemental Fig. 2f) or β -actin (all others) and the analysis was performed again. Furthermore, the reference values were given in all figures showing qPCR data as shown below.

Fig. 1f: **Effect of LIM on ER-stress responsive genes expression**
We redid the qPCR analysis using actin as an internal control.

Minor Comment 3:

Western blot: Quantitative densitometric data should be included in the Figures throughout the manuscript. And molecular mass markers should be shown in the western blots.

Response: In accordance with this comment, quantitative densitometric data were added throughout the manuscript as shown in response to Major comment 4. Molecular mass markers and non-cropped blots will be shown after acceptance.

Minor Comment 4:

In myopic eyes, the axial elongation was mainly due to an increase in vitreous chamber depth. Please provide the vitreous chamber depth data throughout the manuscript.

Response: In accordance with this comment, we added the vitreous chamber depth data throughout the manuscript. When measuring the axial length of the eye, our research team measures the length from the apex of the cornea to the optic nerve papilla in order to always measure the same place. Since OCT images near the optic nerve papilla are blurred and the boundary between the vitreous and retina is unclear, we measured it as VCD plus retinal thickness (RT). The evaluation of VCD+RT was based on the change in VCD+RT before and after the LIM period, not intraocular difference. Because some compounds induced myopia without negative lens wearing, taking the left-right difference could be misinterpreted as suppressing myopia in such situations, for example in Fig. 3b and c. The following data is an example of data that measures VCD+RT (Supplemental Fig. 6f).

f.

Supplemental Fig. 6f: Effect of inhibitor mixture on change in VCD+RT

As shown in DMSO-NL group (represent change with normal eye growth), VCD + RT shortens with growth, myopia induction attenuates the change.

Clarifications needed:

1) Figures 1. Panels f and g are only described in the legend, but the actual data is not shown.

Response: We regret this oversight. Throughout the paper, we made sure to check strictly for such mistakes.

2) Figures 1c. It is difficult to see the rough endoplasmic reticulum (RER) in the control eyes.

Perhaps a higher magnification image is needed (refer to J Ophthalmol. 2020 Jun 10;2020: PMID 3264525.)?.

Response: TEM images were again acquired and to make the endoplasmic reticulum easier to understand, an arrowhead has been added to the corresponding region.

3) The figures do not follow the same sequence as presented in the text. For example, Figure 2f-h (line 91) appeared before Figure 2d-e (line 104). Other sections are similarly disordered and confusing.

Response: We regret this oversight. Throughout the paper, we made sure to check strictly for such mistakes.

Response to Reviewer #2

We wish to express our strong appreciation to you for your insightful comments on our paper. We feel the comments have helped us make significant improvements. Based on your comments, we have performed additional experiments and additional analysis as follows:

Major Comment 1:

This reviewer cannot recognize the novelty of this study because Zhu C. et al. have already reported that ER stress involves in scleral remodeling in a Guinea pig model of form-deprivation myopia and that 4-PBA has the potential to decrease COL1A1 in Guinea pig scleral fibroblasts treated tunicamycin. [Zhu C, Chen Q, Yuan Y, Li M, Ke B. Endoplasmic Reticulum Stress Regulates Scleral Remodeling in a Guinea Pig Model of Form-Deprivation Myopia. *J Ophthalmol.* 2020;2020:3264525. Published 2020 Jun 10. doi:10.1155/2020/3264525.] This reviewer asks the authors to revise the abstract to convey the novelty.

Response: Thank you for reading and commenting on our paper. In Zhu C. et al., they reported that myopia induced by form-deprivation induces endoplasmic reticulum stress in the sclera, but whether it was the result or the cause of the progression of myopia was not shown. In other words, it's just phenomenology. So, they did not show “involvement” of ER stress and myopia. In this regard, we have found that ER stress causes the onset and progression of myopia using various pharmacological methods and gene editing techniques and have also clarified the involvement of PERK and ATF6 pathways. Furthermore, we have also verified the drug kinetics in different dosage forms of 4-PBA (instillation and intraperitoneal injection) and found that 4-PBA reaches the sclera reliably and that instillation can suppress myopia safely and effectively without any side effects. In addition to the novelty of these findings, our strength is that we have demonstrated these findings *in vivo*. In according to this comment, we have revised the abstract as follows:

“We demonstrated that scleral ER stress and PERK/ATF6 pathway controls axial elongation during the myopia development *in vivo* model and 4-PBA eyedrop is a promising drug for myopia suppression/treatment”

Major Comment 2:

In this study, the authors discovered the involvement of ER stress. However, the reason why the authors focused on ER stress is lack. This reviewer hopes to add the description. For example, are there any reports that ER stress (such as ATF6 or PERK activation) would be observed in the sclera of pathological myopia patients?

Response: We focused on ER stress because we found endoplasmic reticulum expansion as a result of detailed observation of myopic sclera using electron microscopy, and there was no supporting evidence for this in previous studies at that time. However, there are a number of reports that ER stress and the UPR pathway are involved in organ size control and as you mentioned above, Zhou et al. reported that endoplasmic reticulum stress occurs in myopic sclera. We have added the bibliographic information to the introduction as below.

“As axial elongation is a change in the organ size, which is probably achieved by remodelling of ECM in the sclera, we hypothesized that scleral ER stress participates in the onset and progression of axial elongation during myopia development. In fact, there have been recent reports indicating that ER stress occurs in the myopic sclera using a form-deprivation myopia model in guinea pig”

Major Comment 3:

In Figure 1c, this reviewer recognized the dilated endoplasmic reticulum (ER) in LIM eye. However, this reviewer did not understand the ER in the control eye. To compare the normal ER with the pathological one, the authors had better show the ER in the control eye using the mark such as asterisk.

Response: In accordance with this comment, the endoplasmic reticulum is now marked with arrowheads for easier understanding as below.

Fig. 1c: TEM images of scleral fibroblasts in normal (NL) and myopic (-30D) sclera
We changed to new images and added arrowheads indicating rough endoplasmic reticulum.

Major Comment 4:

In pages 5, lines 63-65 (Figure 1d); This reviewer cannot admit the description because this reviewer cannot see the increasing of ER-stress related proteins in the LIM group based on only their typical images. Considering the importance of Figure 1d in this study, this reviewer asks the authors to replace the typical images to support the authors' conclusion and to quantify the immunoblots of Figure 1d. In addition, the immunoblots of cleaved ATF6 are pale color. Please improve the typical images.

Response: In accordance with this comment, we changed the image of the blots to a more typical one, and results of quantitative densitometric measurements using image J are also shown in Fig. 1e.

Fig. 1d and e: **Effect of LIM on ORE, eIF2 and ATF6 activation in the sclera.** We redid LIM experiments with age-matched control and performed western blotting and quantitative densitometric measurements.

Major Comment 5:

In Figure 2, the authors showed the potential of 4-PBA to inhibit LIM development by intraperitoneal injection and eyedrop. However, this reviewer doubts the following issues: first, were 4-PBA actually delivered to sclera by intraperitoneal injection in Figure 2b? Did 4-PBA intraperitoneal administration show some adverse effects? Second, how about the influence of 4-PBA eyedrop on anterior segment such as the thinning of lens? Third, does 4-PBA decrease the proteins related PERK and ATF6 pathway? The authors should evaluate whether 4-PBA might decrease the expression of cleaved ATF6 and PERK activation.

Response: Regarding the first issue, we investigated whether 4-PBA and its metabolite, phenylacetic acid (PAA), could be detected in the ocular tissues of mice treated with intraperitoneal or ocular administration of 4-PBA for one week by LC-MS/MS. As a result, 4-PBA was detected in the sclera by both administration methods, indicating that the administered 4-PBA can actually reach and act on the sclera. However, considering the total amount of 4-PBA and PAA detected (as shown in below Supplemental Table 1 and 2), it is thought that the highest amount of 4-PBA reached the retina and was immediately metabolized and converted to PAA because the retina is a metabolically active tissue. The large amount of PAA in the retina may have adversely affected the visual function assessed by ERG (as shown in Supplemental Fig. 4 a-d).

Table 1 Concentrations of 4-Phenylbutyric Acid in mouse ocular tissues after 4-Phenylbutyric Acid treatment

Treatment	Sample	Concentration of 4-Phenylbutyric Acid (ng/g)				
		1	2	3	Mean	SD
4-Phenylbutyric Acid	Retina	209	221	165	198	29
	Choroid	327	520	208	352	157
	Sclera	829	691	117	546	378
PBS	Retina	BLQ	BLQ	BLQ	BLQ	NC
	Choroid	BLQ	BLQ	BLQ	BLQ	NC
	Sclera	BLQ	BLQ	BLQ	BLQ	NC

BLQ : Below the lower limit of quantification

Table 2 Concentrations of Phenylacetic Acid in mouse ocular tissues after 4-Phenylbutyric Acid treatment

Treatment	Sample	Concentration of Phenylacetic Acid (ng/g)				
		1	2	3	Mean	SD
4-Phenylbutyric Acid	Retina	34100	33900	30900	33000	1800
	Choroid	7780	5120	6040	6310	1350
	Sclera	1610	1480	1500	1530	70
PBS	Retina	BLQ	BLQ	BLQ	BLQ	NC
	Choroid	BLQ	BLQ	BLQ	BLQ	NC
	Sclera	BLQ	BLQ	BLQ	BLQ	NC

BLQ : Below the lower limit of quantification

NC: Not calculated

Supplemental Table 1 and 2: Amount of 4-PBA and PAA in each eye tissue after 1 week of 4-PBA intraperitoneal administration

Administered 4-PBA delivered to sclera.

Supplemental Fig.4 a-d: Effect of i.p. 4-PBA injection on retinal function.

Intraperitoneal 4-PBA injection had harmful effect on the retina determined by ERG measurement.

Regarding the second comment about the influence of 4-PBA eyedrop on the anterior segment, we added the results of measuring lens thickness and evaluating mucin expression in the conjunctiva in the 4-PBA eyedrop experiment because ER stress and/or 4-PBA is associated with mucin production in the conjunctiva and lens homeostasis (Coursey et al, *Am J Pathol* **186**, 1547-1558, 2016; Zhou et al, *Mol Med Rep* **21**, 173-180, 2020). The lenses tended to thicken with myopia induction, but 4-PBA thinned the lenses with statistical significance (Supplemental Fig. 4k). This result is consistent with the fact that myopia is suppressed. And, 4-PBA had no effect against mucin expression in the conjunctiva (Supplemental Fig. 4i).

Supplemental Fig. 4: Effect of 4-PBA instillation on retinal anterior segment
i: Mucin expression in the conjunctiva
k: lens thickness

To determine whether 4-PBA inhibits the activity of PERK and ATF6 induced by LIM, we performed Western blotting and quantitative analysis of the blots (Fig. 2a, b, e, f). These results demonstrated that 4-PBA administration, both intraperitoneally and eyedrop, suppressed eIF2 and ATF6 activation as shown below.

Fig. 2: Effect of 4-PBA administration on activation of IRE1, PERK and ATF6 pathways. a and b: results of i. p. injection e and f: results of instillation

4-PBA suppressed LIM-induced activation UPR pathways.

Major Comment 6:

In Figure 1e and 2a, what is the method of statistical analysis? Is it one-sided test or two-sided test? This reviewer doubts that some ER stress markers have significant differences considering the statistical variability. In addition, the asterisks, which means significantly differences, should be marked on the column of 4-PBA in Figure 2a.

Response: In the previous version of manuscript, we performed one-way statistical processing, but in the revised manuscript, we performed a student’s t-test for Fig. 1e (move to Fig. 1f) and a two-sided test for Fig. 2a (move to Supplemental Fig. 3a).

Major Comment 7:

The authors demonstrated that GSK2656157 (PERK inhibitor) plus nefinavir (ATF6 inhibitor) co-instillation completely inhibited LIM development (Figure 3c and 3d). On the other hand, the authors did not evaluate whether GSK2606414 (PERK inhibitor) plus Ceapin-7 (ATF6 inhibitor) would inhibit LIM development in Extended Data Figure 6. This reviewer asks the authors to evaluate whether GSK2606414 plus Ceapin-7 also have the potential to inhibit LIM development. This data might be important to confirm that PERK/ATF6 pathway involves in LIM development.

Response: In accordance with this comment, we additionally performed co-instillation experiments using 4μ8C, GSK2606414, Ceapin-7 and the results are shown in Supplemental Fig. 6g and h as shown below.

Supplemental Fig. 6: Effect of co-institution of other inhibitors on axial length and refraction.

Major Comment 8:

The author demonstrated that ATF6 and PERK would involve in LIM pathology by pharmacological approach. However, considering of many compounds has some unexpected effects, this reviewer suggests the authors to demonstrate them by another approach. For example, using shRNA or siRNA of ATF6 and PERK might be important to confirm that ATF6 and PERK might involve in LIM pathology.

Response: In accordance with this comment, we constructed Adeno-Associated Virus vectors for knockdown of PERK (Eif2ak3) and ATF6 by the CRISPR/Cas9 system and injected them into the sub-tenon's capsule to knockdown these genes in the sclera and then evaluated myopia development by LIM. The results were not completely consistent with the results of the pharmacological experiments, but they were similar, suggesting the involvement of both PERK and ATF6 in myopia progression (Fig. 3h-j, as shown below).

Fig. 3: Effect of knockdown of PERK (Eif2ak3) and Atf6 using AAV-delivered CRISPR/Cas9 system on gene expression (h), change in axial length by LIM (i) and change in refraction (j).

Sub-tenon's injection of Cas9/guide RNA-packaging AAV could decrease PERK or ATF6 gene expression. ATF6 alone or both ATF6 and PERK knockdown suppressed myopic changes.

Major Comment 9:

In Extended Figure 2e (and 2f), the authors described that cell death was not occurred in LIM.

However, the authors supplied only the typical images. Therefore, this reviewer asks the authors to

quantify the TUNEL-positive cells.

Response: In accordance with this comment, we have quantified the TUNEL-positive cells and added the result in Supplemental Fig. 2d.

Major Comment 10:

The involvement of ER stress on collagen expression is unclear in this study. TGF β -SMAD signaling pathway is one of the major pathways to express collagen. Therefore, this reviewer asks the authors to evaluate SMAD activation on the sclera of LIM group and 4-PBA treated group.

Response: We appreciate your very constructive comments. Based on previous studies, we evaluated the phosphorylation of SMAD1 by LIM and administration of 4-PBA. We found that phosphorylation of SMAD1 was lower in myopia-induced eyes with or without 4-PBA, suggesting that SMAD1 is not involved in myopia-induced scleral remodeling and its suppression by 4-PBA (Supplemental Fig. 7c, as shown below). One possibility for the molecular mechanism linking ER stress and collagen expression is the involvement of both PERK and ATF6 pathways. By treating scleral fibroblasts with Tm for 6 hours, the expression of collagen 1 is markedly reduced. In this occasion, STF (IRE1 inhibitor), GSK (PERK inhibitor) and NFV (ATF6 inhibitor) were added alone or in combination, the decrease in collagen 1 expression was completely suppressed in the presence of both GSK and NFV (Supplemental Fig. 7d, as shown below). This result is consistent with the fact that both GSK and NFV suppressed myopia (Fig. 3f and g), and their involvement is strongly considered.

Major Comment 11:

In this LIM model, posterior staphyloma, which causes pathologic myopia, is observed? If so, is posterior staphyloma decreased by 4-PBA treatment? Considering of formation of posterior staphyloma is a key component of a spectrum of vision-threatening myopic maculopathies, this reviewer is interested in it.

Response: Ocular morphology after LIM was observed by SD-OCT and HE-stained paraffin sections, but no staphyloma was observed. We believe that our myopia model is a model of high myopia, but not enough to induce pathological ocular complications.

Minor Comment 1:

In the legend of Figure 1, the authors described Figure 1f and 1g. However, this reviewer could not find Figure 1f and 1g. The authors should address this issue.

Response: We regret this oversight. Throughout the paper, we made sure to check strictly for such mistakes.

Minor Comment 2:

Please spell out “NL” in Figure 1 legends.

Response: In accordance with this comment, we spelled out “NL” as “No Lens”.

Minor Comment 3:

Please magnify the font size of Figure 1e.

Response: In accordance with this comment, the Figure has been modified to allow larger text, and the text has been enlarged to a readable size.

Minor Comment 4:

The authors should revise the following misprints:

In pages 6, lines 91; “Fig. 2f”→”Fig. 2d”

In pages 6, lines 95; “Fig. 2g and h”→” Fig. 2e and f”

In pages 7, lines 103-104; “Fig. 2d and e”→“Fig. 2g and h”

Response: We regret this oversight. Throughout the paper, we made sure to check strictly for such mistakes.

Minor Comment 5:

In Figure 2g and 2h, what for animals were 0.2% and 2% 4-PBA applied? LIM treated group or Non-LIM treated group? Please add the description.

Response: In accordance with this comment, we added the description into Materials & Methods section and into Figure legend as shown below.

“Ocular eyedrop administration of 4-PBA (2% solution in PBS) during LIM period (to both eyes) suppresses LIM-induced activation in UPR branch determined by Western blotting in sclerae”

Minor Comment 6:

In this study, the authors used many kinds of compounds to regulate ER-stress. Therefore, this reviewer recommends the authors to make the table showing their target to enable to read easily such as STF083010: IRE1 inhibitor.

Response: In accordance with this comment, we made a table and added it as Supplemental Table 8 as shown below. This should outline the methodology more clearly.

Agonists/Antagonists	Abbreviations in this paper	Action and mechanism
4-phenylbutyric Acid	4-PBA	Acts as a chemical chaperone to aid in protein folding and consequently reduce ER stress
Tauroursodeoxycholic Acid	TUDCA	Chemical chaperone which can inhibit unfold protein response dysfunction and ameliorate ER stress
Tunicamycin	Tm	Inhibits glycoprotein biosynthesis in the ER which result in accumulation of misfolding protein, followed by induction of ER stress
Thapsigargin	TG	Inhibits sarco/endoplasmic reticulum Ca ²⁺ -ATPase which caruse ER stress
STF083010	STF or S	IRE1 inhibitor through blocking IRE1 endonuclease activity
4 μ 8C	4 μ 8C or 4	IRE1 inhibitor through blocking substrate access to the active site of IRE1
GSK2656157	GSK or G	PERK inhibitor thorough competing with ATP
GSK2606414	GSK2606414 or G	The first PERK inhibitors
Nerfinavir	NFV or N	ATF6 inhibitor through blocking Site-2 proteinase cleavage of ATF6
Ceapin-7	Ceapin or CP	ATF6 inhibitor through preventing transport of ATF6a to the Golgi apparatus
CCT020312	CCT	A selective PERK activator which elicits eIF2 phosphorylation
AA147	AA	A selective ATF6 activator

Minor Comment 7:

In pages 11, lines 174; Extended Data Fig. 2f is lack in this manuscript.

Response: We regret this oversight. Throughout the paper, we made sure to check strictly for such mistakes.

Minor Comment 8:

In Extended Data Figure 5b: the vertical line meaning Δ Axial length is hidden.

Response: In accordance with this comment, we revised the graph as below.

Minor Comment 8:

In Extended Data Figure 5h: Please modify “p=0114”.

Response: In accordance with this comment, we modified p value in Extended Data Figure 5h as below.

Response to Reviewer #3

We wish to express our strong appreciation to you for your insightful comments on our paper. We feel the comments have helped us significantly make improvements. Based on your comments, we have performed additional experiments and additional analysis as follows:

Comment 1:

In Fig 1c, they showed the enlarged rough ER in sclera of LIM eye. Is it shown only in sclera or, in the other region of eyeball layer? Please add the result of whether LIM induced-enlarged ER is only shown in sclera or the other layer. The EM images are very difficult to interpret subcellular organelles because they appear out-of-focus or are too low-magnification.

Response: We appreciate your comment on this point. We have additionally shown that ER stress does not occur in the retinal layer determined by Western blotting and qPCR (Supplemental Fig. 2f, g). This result suggests that there are few cells in the retina that show endoplasmic reticulum expansion. In support of this consideration, no expansion of the endoplasmic reticulum was observed, at least in RPE cells (Supplemental Fig. 2h). And the endoplasmic reticulum is now marked with arrowheads for easier understanding (Fig. 1c).

Supplemental Figure. 2

Figure. 1

Fig. 1c: **TEM images of scleral fibroblasts in normal (NL) and myopic (-30D) sclera**

We changed to new images and added allowheads indicating rough endoplasmic reticulum.

Comment 2:

Please add detail information of antibody catalog number in method section.

Response: In accordance with this comment, we have provided the catalog numbers for the antibodies in method section.

Comment 3:

In Fig 1d, it is hard to see the difference of ER stress protein level, phospho-IRE1, not like their manuscript description. Please add the statistical analysis graph by normalizing with suitable control protein. In addition, phospho-eIF2a seems to be decreased by normalization of eIF2a. Please discuss about the point. Last, weight markers are needed to confirm bands are correct size.

Response: In accordance with this comment, we re-performed LIM experiments with age-matched control followed by Western blotting and added the statistical analysis graph (Fig. 1d and e, as shown below). As a result, all 3 pathways were activated in minus-lens wearing sclerae compared to age-match control or No Lens (NL) control. The text was also proofread accordingly. Regarding the molecular weight markers, it is difficult to describe the markers because all the figures are very tight, but after the review we will post the uncropped blots along with the molecular weight markers, so we hope you will refer to them.

Fig. 1d and e: **Effect of LIM on ORE, eIF2 and ATF6 activation in the sclera.** We redid LIM experiments with age-matched control and performed western blotting and quantitative densitometric measurements.

Comment 4:

In Fig 1e, please add information of what statistical analysis they used is to show the significance and how the significance is calculated by comparison of what.

Response: In accordance with this comment, we added the information about statistical analysis and comparison in the figure legend.

Comment 5:

In Fig 2a, they showed 4-PBA suppressing UPR gene expression. However, it is not clear whether it is for 1 or 3 weeks. Please add the detail information of experiment time frame.

Response: In accordance with this comment, we added the experiment time frame in the figure legend (the figure was changed to Supplemental Fig. 3a). The experimental period for that experiment was three weeks.

Comment 6:

Fig 2 a,b,g,h seem to be replication with Extended Data Fig 4 a,b,g,h. Please replace the Fig 2 a,b,g,h by Extended Data Fig 4 a,b,g,h. Otherwise, please address why you displayed like that.

Response: In accordance with this comment, we deleted Fig. 2a, b, g, h and replaced them with Extended Data Fig. 4a, b, g, h (The revised version is Fig. 2c, d, g, h.).

Comment 7:

In Fig 2d, they need to mention what kind of chop expression it is. Is it RNA or protein? Also, they need to describe what LIM time period of sample they tested.

Response: Fig. 2d shows the results of qPCR analysis of CHOP mRNA in the sclera 6 hours, 1 day, 3 days, and 7 days after a single dose of tunicamycin (50 µg/mL) instillation. We added the information in the figure legend in accordance with your comment (Fig. 2d moved to Fig. 2j in the revised manuscript).

Comment 8:

Please correct figure numbering (Fig.2f, 2g, 2h, Extended Data 4b~f, 4g~k, and so on). There is no Extended Data 4k on figure, not like manuscript description.

Response: We regret this oversight. Throughout the paper, we made sure to check strictly for such mistakes.

Comment 9:

In Fig 3 a~d, I am doubt if their data interpretation of statistic comparisons is reasonable. They should compare the result of each compound in that of same model, or each chemical in same mouse, not like comparing with only NL/DMSO. To conclude the effect of ER stress inhibitors, they should redo statistic analysis comparing with an eligible control.

Response: The original statistical analysis was done using one-way analysis of variance, however we learned the method is not correct from your comments, we redid statistical analysis using Generalized Estimating Equations to make comparisons between left and right and between compounds. As a result, as in the previous version, only the DMSO group and the group treated with

STF alone showed differences between the left and right eyes between the same compounds, and only between DMSO and G+N showed differences in the comparison between the right eyes (LIM eyes). We also redid the statistical process throughout the manuscript.

Comment 10:

They need to add primer information for qPCR. They missed the Extended Table 1 for the information, not like their description on method section.

Response: We apologize for the missing table. Tables with primer information have been added in Extended Table 1-3.

Comment 11:

In Fig 3e, f, and g, they should mention concentration information of each chemical.

Response: Fig. 3e has been removed because the effect of mixed doses of inhibitors has been evaluated by Western blotting. This experiment was performed at 100 μ M of each inhibitor, and this information is provided in the Methods section and figure legend. For Fig. 3f and g (moved to Fig. 3l and m), the dose concentration is indicated on the label on the horizontal axis as below.

Fig. 3l and m: Effect of different concentration of CCT020312 (l) or AA147 (m) on ER-stress responsible gene expression

Comment 12:

In Fig 3e~m, they should mention what model they compared between, NL or -30D.

Response: The experiments shown in Fig. 3 e~m were the comparison between mice that were instilled with each compound in both eyes and mice that received PBS containing the same amount of DMSO as a control. One-way ANOVA was used for comparison between the four groups in Fig. 3f, g (moved to Fig. 3l and m) and the comparison between the two groups in Fig. 3h-m (moved to Fig. 3n-p) was made using student's t-test.

Comment 13:

In Fig 4, they use the word, moderate ER stress, in their Tm-NIH 3T3 cell culture model. But they did not show how they determined the moderate concentration of ER stress in mice model. Using Tm dose-dependent curve test, please add the result of Tm-induced ER stress in LIM model

Response: We used the word “moderate” to mean that the endoplasmic reticulum stress was not strong enough to cause apoptosis, but as you said, the definition is ambiguous, so we revised the use of that word and the expression in the text has been changed to "concentration that does not cause apoptosis. We also evaluated scleral ER stress by Tm instillation in Western blotting and added the results in Fig. 2i as shown below.”

Fig. 2i: Effect of tunicamycin (Tm) instillation on IRE1, eIF2 and ATF6 activation in the sclera.

REVIEWER COMMENTS

Reviewer #1 (Remarks to the Author):

The authors have conducted new experiments (age-matched control group and the Cas9-genetic manipulation) and most comments have been addressed satisfactorily. I would like to thank the authors for adding the quantitative densitometric data of immunoblots. They have made the thesis much more credible now.

Nevertheless, I still have some old and some additional concerns.

1. Line 32, 34 and through the manuscript: Based on the current data, the authors have not showed that the ER-stress was specifically occurred in scleral fibroblasts. Please delete the "fibroblast".
2. line 60-65: The reviewer suggests the author to reverse the sequence of these two sentences.
3. Line 147-148: The authors showed the ocular administration of 4-PBA affected the thickness of Lens, which will affect the refraction in mice. Please discuss this concern.
4. Line 155: The current data did not support the conclusion that targeting ER stress could "prevent" myopia development.
5. Line 213: This method results in gene knock-down in the whole sclera, rather than selectively in scleral fibroblasts.
6. Line 211-288: It would be important to determine the effect of the AAV-mediated gene manipulation on the normal refraction development without LIM treatment.
7. AAV8 injection: i) The authors have not provided data to demonstrate that 5 ul of AAV injection could results in infection of the global sclera. This conclusion would be considerably strengthened if the authors could localize the site of virally delivered in the sclera. ii) If this procedure will affect other ocular tissues? iii) The serotype of the AAV8 should be provided in the manuscript.
8. BIP IHC and TUNEL assay: The IHC quantification needs to be better described. It is still extremely hard to see the positive signal in most of the images (Supplemental Fig 2a, and c, Figure S7a). A higher magnification image should be included. In Supplemental Fig 2c and d, the TUNEL staining is merged with DAPI, which makes the TUNEL is hard to identify. In addition, it's incredible that 10% of the scleral cells are under apoptosis (Supplemental Fig 2d). This raises some doubts.
9. Tunicamycin concentration: The authors have not provided the convincing reason why 200ng/ml was employed in mice. This concentration was not included in the in vitro experiment in Supplemental Fig 7. In addition, this in vitro experiment-derived concentration should not be directly translated to the corresponding dosage for animal experiments.
10. TGF β -smad: Smad2 and Smad3 are the two major downstream regulator that promote TGF β -mediated tissue fibrosis. The authors need to explain why the Smad1/5, but not the Smad2/3, was included in the current study.

Reviewer #2 (Remarks to the Author):

The authors addressed this reviewer's revise comments, and confirmed the involvement of ER stress and pathological myopia. However, there are still some questions in this study. In your response #5, the authors claimed "The large amount of PAA in the retina may have adversely affected the visual function assessed by ERG", but this reviewer cannot admit this

description because the author did not directly show the adversely effect of PAA. Therefore, this reviewer asks the authors to show the evidence of the adverse effect of PAA by additional experiment or comment. In addition, in the supplemental Figure 4a-d of revised manuscript, ERG data showed the harmful effect of 4-PBA on retinal function. Therefore, this reviewer doubts the data such as Figure 2a-d because it is unknown whether 4-PBA modulated axial elongation via therapeutic effect such as reducing ER stress or via harmful effect on retinal function. Please add the comment for this issue.

Reviewer #3 (Remarks to the Author):

Review overview: Dr. Shin-ichi and colleagues revised data of PERK/ATF6 controlling axial elongation in myopia pathology, and 4-PBA attenuating LIM-derived ER stress and remodeling gene expression related to scleral collagen. Interestingly, they also updated that the administration method of 4-PBA instillation is more effective than that of i.p. injection. Although the authors updated some data, there are still overinterpretation and errors in data analysis, sloppy description of figure legends, irrational statistical interpretation, errors (wrong numberings of Table), and rough experimental design. Overall, the manuscript needs appropriate experimental controls in order to interpret the data and support their conclusions, significant revision, and editing.

1. In Fig 1d, p-eIF2 is significantly increased in -30D. The phosphorylation of eIF2 can be modulated by ISR pathway (GCN2/PKR/HRI) without PERK (Mol Biol Cell. 2005 Dec;16(12):5493-501; Molecular Cell Vol 6, Issue 5, November 2000, Pages 1099-1108). Please check up PERK level or phospho-PERK level in LIM with a suitable control.
2. In Fig 2a, 2i, the WB figures of p-eIF2a and ATF6-N do not represent the difference of statistic graph. Please advise about the discrepancy between the blots and quantification and update with better results like Fig 2e.
3. On line 124, 125, 126, the Supplemental Table numberings are wrong. Please correct the table numberings.
4. In Supplemental Fig 3d and 3h figure legend, it needs to be explained or added the comparison bar where the significance is coming from.
5. Please keep the defined word consistent. For example, Extended Table > Supplemental Table.
6. In Fig 2i~2l and Supplemental Fig 3i~3n, the authors confirmed that ER stress compounds (TG/Tm) induces similar effect of LIM-derived myopia. The reviewer has a question if the authors tested the effect of 4-PBA instillation in TG/TM induced myopia. If so, please add discussion about the result.
7. In Fig 3b, 3c, 3f, 3g, Supplemental Fig 6b~6i, the statistic comparisons are not well designed and performed. Like the Fig 2g, 2h, they should compare the result of each compound with that of same experimental condition, not like comparing with only NL/DMSO. For example, they should compare -30D/DMSO with -30D/STF, -30D/GSK, or -30D/NFV. Then, please update the results on line 180~182, 192~196. Or please delete -30D data.
8. In Fig 3h, to confirm the result of PERK and ATF6 pharmacological intervention, they generated SaCas9-derived PERK and ATF6 knock-down model and tested the myopia development. Although it is impressive approach, they did not validate the model carefully. The error bar of control appears highly variable. Please confirm protein expression levels using western blot assay. Or please provide the other validation data such as DNA, RNA sequencing, or gene set enrichment analysis.
9. In Fig 3i and 3j, the genetic data of ATF6 or ATF6/PERK is not consistent with the

pharmacochemical data of ATF6 and PERK inhibitors. Please discuss the reason.

10. In Fig 3k, please explain incubation time of drug instillation in figure legend. And please update p-eIF2, ATF6-P blots which are hard to see the difference.

Response to Reviewer #1

Introductory comments:

The authors have conducted new experiments (age-matched control group and the Cas9-genetic manipulation) and most comments have been addressed satisfactorily. I would like to thank the authors for adding the quantitative densitometric data of immunoblots. They have made the thesis much more credible now.

Nevertheless, I still have some old and some additional concerns.

- Your previous comments were very important, and we wish to express our strong appreciation to you for your insightful comments, which have strengthened the results of our paper by conducting new experiments. Based on your additional comments, we have performed additional experiments and additional analysis as follows:

Comment 1:

Line 32, 34 and through the manuscript: Based on the current data, the authors have not showed that the ER-stress was specifically occurred in scleral fibroblasts. Please delete the “fibroblast”.

Response: We appreciate your comment on this point. In accordance with this comment, we deleted the “fibroblast” in above two lines.

Comment 2:

line 60-65: The reviewer suggests the author to reverse the sequence of these two sentences.

Response: In accordance with this comment, we switched the order of the two sentences.

Major Comment 3:

Line 147-148: The authors showed the ocular administration of 4-PBA affected the thickness of Lens, which will affect the refraction in mice. Please discuss this concern.

Response: In accordance with this comment, we added the discussion about the concern in Results section as shown below,

“Lens thickening due to minus lens wearing is thought to be a compensatory effect for focus shift, but since myopia was suppressed by 4-PBA eye drops, lens thickening due to minus lens wearing did not occur in the 4-PBA instillation group.”

Comment 4:

Line 155: The current data did not support the conclusion that targeting ER stress could “prevent” myopia development.

Response: In accordance with these comments, we removed the word "prevent".

Comment 5:

Line 213: This method results in gene knock-down in the whole sclera, rather than selectively in scleral fibroblasts.

Response: In accordance with this comment, we have corrected the wording when referring to knockdown experiments.

Comment 6:

Line 211-288: It would be important to determine the effect of the AAV-mediated gene manipulation on the normal refraction development without LIM treatment.

Response: We used AAV-mediated gene manipulation both eyes, the right eye is mounted with a minus lens to induce myopia, while the left eye is not mounted with a lens and is controlled against the right eye. Therefore, we thought that a comparison of the data for the left eye in each group answers this comment, and a summary of the data for the left eye only is shown below. Since the changes in axial length and refraction occurred only when a particular gene was expressed, it is thought that genetic manipulation by AAV itself does not affect the normal refractive situation.

Comment 7:

AAV8 injection: i) The authors have not provided data to demonstrate that 5 ul of AAV injection could results in infection of the global sclera. This conclusion would be considerably strengthened if the authors could localize the site of virally delivered in the sclera. ii) If this procedure will affect other ocular tissues? iii) The serotype of the AAV8 should be provided in the manuscript.

Response: i) To verify whether sub-tenon's injection of AAV(DJ) causes scleral infection, AAV(DJ)-EGFP was administered, and 28 days later, whole-mounts of the sclera were made and

EGFP was observed. The results are shown as below and added in Fig. 4a. EGFP fluorescence was observed in the entire sclera, especially in the posterior region, indicating that sufficient gene transfer can be expected.

a.

Fig. 4a: Distributions of EGFP expression in the scleral whole-mount after 28 days of AAV serotype DJ (AAV-DJ)-EGFP or AAV-DJ-Cas9.

EGFP was apparent in the whole sclera in the sclera of AAV-DJ-EGFP-injected eyes; however, no EGFP was detected in the sclera of AAV-DJ-Cas9-injected eyes

ii) We have observed the layered structure of the retina using SD-OCT, but the results showed no abnormalities in the retinal morphology as shown in Fig 4b and as below.

Fig. 4b: Representative OCT images from each group

No abnormalities in retinal morphology were observed in any of the treatment groups.

iii) The AAV serotype we used in this study is AAV-DJ (type2/type8/type9 chimera), which was developed by Grimm et al. in 2008 (*J Virol*, 2008 Jun;82(12):5887-911). We have described them throughout the paper.

Comment 8:

BIP IHC and TUNEL assay: The IHC quantification needs to be better described. It is still extremely hard to see the positive signal in most of the images (Supplemental Fig 2a, and c, Figure S7a). A higher magnification image should be included. In Supplemental Fig 2c and d, the TUNEL staining is merged with DAPI, which makes the TUNEL is hard to identify. In addition, it's incredible that 10% of the scleral cells are under apoptosis (Supplemental Fig 2d). This raises some doubts.

Response: In accordance with this comment, we replaced BiP IHC image and TUNEL image to a higher magnified image as below. In addition, as you pointed out, the fact that as many as 10% of the cells underwent apoptosis was certainly disconcerting, so we changed the original automatic measurement on the observation software to manual measurement and replaced Supplemental Fig. 2d with the results of the new measurement.

Supplemental Fig. 2 a and c: **Representative BiP IHC images and TUNEL staining images**
The image has been changed to a higher magnified image version.

Supplemental Fig. 8a: **Representative TUNEL staining images of Tm-treated huScF.**
The image has been changed to better visibility.

Comment 9:

Tunicamycin concentration: The authors have not provided the convincing reason why 200ng/ml was employed in mice. This concentration was not included in the in vitro experiment in Supplemental Fig 7. In addition, this in vitro experiment-derived concentration should not be directly translated to the corresponding dosage for animal experiments.

Response: In the previous manuscript, the 200ng/ml data was missing, so Supplemental Fig 7a was changed to a Figure with that data added (Supplemental Fig 8a, and shown above response to Comment 8). The doses of Tm and TG in the animal experiments are not the same as those in the cell experiments, because the doses were decided from those administered to animals in previous studies (Wang et al, *Exp Eye Res*, 2019; Nakamura et al, *PLoS ONE*, 2013). We have cited these two papers in the manuscript.

Comment 10:

TGF β -smad: Smad2 and Smad3 are the two major downstream regulator that promote TGF β -mediated tissue fibrosis. The authors need to explain why the Smad1/5, but not the Smad2/3, was included in the current study.

Response: After receiving the previous comment, we first made a preliminary examination of Smads that are altered by myopia induction, and found that Smad2/3 is not altered by LIM, while Smad1/5 has its phosphorylation level decreased. Therefore, we decided to target Smad1/5. These data were added in Supplemental Fig 8c and shown below. Further reason was that there are reports that Smad1/5 regulate collagen expression (Chen et al, *Life Sci*, 2019; Finnson et al, *Osteoarthritis Cartilage*, 2010; González-Núñez et al, *Biochim Biophys Acta*, 2013), especially collagen 4 (Abe et al, *J Biol Chem*, 2004: a collagen gene whose expression was markedly increased by LIM).

C.

Supplemental Fig. 8c: **Effect of LIM on Smad phosphorylation in sclera.**

LIM induced decrease in SMAD1/5 phosphorylation level in sclera. However, SMAD2/3 phosphorylation level did not change by LIM.

Response to Reviewer #2

Introductory comments:

The authors addressed this reviewer's revise comments, and confirmed the involvement of ER stress and pathological myopia. However, there are still some questions in this study.

- We wish to express our strong appreciation to you for your insightful comments on our paper. We feel the comments have helped us make significant improvements. Based on your additional comments, we have performed additional experiments and additional revised our manuscript as follows:

Comment 1:

In your response #5, the authors claimed "The large amount of PAA in the retina may have adversely affected the visual function assessed by ERG", but this reviewer cannot admit this description because the author did not directly show the adversely effect of PAA.

Therefore, this reviewer asks the authors to show the evidence of the adverse effect of PAA by additional experiment or comment.

Response: Thank you for reading and commenting on our paper. In our country, PAA is designated as a methamphetamine raw material and requires permission from the local government to be handled. We followed the procedure properly, purchased PAA and tried to conduct experiments to assess the effect of PAA administration on retinal function. As with 4-PBA i.p. injection, we planned to administer 2% PAA solution intraperitoneally to the mice once a day, followed by ERG, but as shown in the Kaplan-Meier curve below, all individuals died by day 4 and we were unable to implement the plan. However, this result shows that PAA is a harmful compound and as mentioned in the manuscript, it is known to have negative effects on the brain. Both the brain and the retina are nerve tissue and based on the results of our PAA administration experiment, we think it is reasonable to assume that the accumulation of PAA in the retina might have some negative effects, as seen in the ERG results.

Comment 2:

In addition, in the supplemental Figure 4a-d of revised manuscript, ERG data showed the harmful effect of 4-PBA on retinal function. Therefore, this reviewer doubts the data such as Figure 2a-d because it is unknown whether 4-PBA modulated axial elongation via therapeutic effect such as reducing ER stress or via harmful effect on retinal function. Please add the comment for this issue.

Response: Since no decrease in retinal function was observed after the administration of 4-PBA eye drops (Supplemental Figure 4e-f), but myopia was suppressed (Figure 2g and h), it is unlikely that the decrease in retinal function itself suppresses myopia. In addition, FDM is a model that induces myopia by blocking vision. Since the reduction in visual information induces myopia, it is likely that the reduction in retinal function acts in the direction of inducing myopia. Based on the above, we thought the myopia-inhibiting effect of 4-PBA can be considered separately from the reduction in retinal function.

Response to Reviewer #3

Introductory comments:

Review overview: Dr. Shin-ichi and colleagues revised data of PERK/ATF6 controlling axial elongation in myopia pathology, and 4-PBA attenuating LIM-derived ER stress and remodeling gene expression related to scleral collagen. Interestingly, they also updated that the administration method of 4-PBA instillation is more effective than that of i.p. injection. Although the authors updated some data, there are still overinterpretation and errors in data analysis, sloppy description of figure legends, irrational statistical interpretation, errors (wrong numberings of Table), and rough experimental design. Overall, the manuscript needs appropriate experimental controls in order to interpret the data and support their conclusions, significant revision, and editing.

- We wish to express our strong appreciation to you for your previous insightful comments on our paper. We feel the comments have helped us significantly make improvements. The authors are convinced that your additional comments are constructive for the further improvement of our paper and that this has resulted in a higher quality paper. In response to your advice, we carried out additional experiments and analyses as follows:

Comment 1:

In Fig 1d, p-eIF2 is significantly increased in -30D. The phosphorylation of eIF2 can be modulated by ISR pathway (GCN2/PKR/HRI) without PERK (Mol Biol Cell. 2005 Dec;16(12):5493-501; Molecular Cell Vol 6, Issue 5, November 2000, Pages 1099-1108). Please check up PERK level or phospho-PERK level in LIM with a suitable control.

Response: We appreciate your comment on this point. In accordance with this comment, we have performed Western blotting of phosphorylated PERK and total PERK and added the results to Fig. 1d and e as below

Fig. 1d and e: Effect of LIM on IRE, PERK, eIF2 and ATF6 activation in the sclera. We performed Western blotting using PERK (phospho- and total) antibodies and quantitative densitometric measurements.

Comment 2:

In Fig 2a, 2i, the WB figures of p-eIF2a and ATF6-N do not represent the difference of statistic graph. Please advise about the discrepancy between the blots and quantification and update with better results like Fig 2e.

Response: When quantifying the blot, the graph shows the result of dividing the value of phosphorylated eIF2 (top panel) by the value of total eIF2 amount (bottom panel) for eIF2, and the result of dividing the value of ATF6-N (bottom panel) by ATF6-P (top panel) for ATF6. Therefore, there should be no discrepancy between the blot pattern and the trend of the graph.

In accordance with this comment, we have changed the images of eIF2 and ATF6 in Fig. 2a and eIF2 and ATF6 in Fig. 2i to those of the other blots.

Fig.2a: Effect of LIM and systemically 4-PBA administration on UPR components.

Fig. 2i: Effect of Tm instillation on UPR components

We replaced eIF2 and ATF6 blots to the other blots.

Comment 3:

On line 124, 125, 126, the Supplemental Table numberings are wrong. Please correct the table numberings.

Response: We regret this oversight. Throughout the paper, we made sure to check strictly for such mistakes.

Comment 4:

In Supplemental Fig 3d and 3h figure legend, it needs to be explained or added the comparison bar where the significance is coming from.

Response: In accordance with this comment, we added the comparison bar in Supplemental Fig 3d and 3h as below.

Comment 5:

Please keep the defined word consistent. For example, Extended Table > Supplemental Table.

Response: We regret this oversight. Throughout the paper, we made sure to check strictly for such mistakes.

Comment 6:

In Fig 2i~2l and Supplemental Fig 3i~3n, the authors confirmed that ER stress compounds (TG/Tm) induces similar effect of LIM-derived myopia. The reviewer has a question if the authors tested the effect of 4-PBA instillation in TG/TM induced myopia. If so, please add discussion about the result.

Response: In the previous manuscript, we had not tested the effect of PBA on TM/TG-induced myopia. But, we thought that doing this experiment would strengthen our results, so we did an additional experiment. The following results were obtained, and they have been added to Supplemental Fig 3m and n, and new text has been added to the Results and Discussion sections based on these results.

Supplemental Fig.3m, n: **Effect of 4-PBA on Tm/TG-induced myopia**
 After Tm/TG instillation, 4-PBA was topically applied once a day (7 days). 4-PBA also suppressed both Tm/TG-induced myopic changes.

Comment 7:

In Fig 3b, 3c, 3f, 3g, Supplemental Fig 6b~6i, the statistic comparisons are not well designed and performed. Like the Fig 2g, 2h, they should compare the result of each compound with that of same experimental condition, not like comparing with only NL/DMSO. For example, they should compare -30D/DMSO with -30D/STF, -30D/GSK, or -30D/NFV. Then, please update the results on line 180~182, 192~196. Or please delete -30D data.

Response: We appreciate your comment on this point. In order to make appropriate statistical comparisons, the data (Fig 3b, c, f, g and Supplemental Fig 6b~i) were divided into two (NL or minus 30D lens-wearing eyes) or four (DMSO, STF, GSK or NFV applied mouse) groups, depending on the treatment or experimental conditions, and statistical analysis was performed within each group. As an example, the analysis and figure of STF, GSK and NFV eye drop experiment are shown below. To assess the effect of each inhibitor on myopia development, ANOVA was used to compare between NL/DMSO, NL/STF, NL/GSK, NL/NFV or -30D/DMSO and -30D/STF, -30D/GSK, -30D/NFV, and the results are shown in Figure 3b and c. Furthermore, to evaluate the effect of minus lens-wearing in each drug applied mouse, student t-test was used to compare between NL/DMSO and -30D/DMSO, NL/STF and -30D/DMSO, NL/GSK and -30D/GSK, or NL/NFV and -30D/NFV, which the results are shown in Supplemental Figure 6a-d. Based on the results of these comparisons, the following changes were made to the text to mention the possible effects of drugs and negative lenses on myopia.

“The axial length, refraction and VCD + RT shortening were comparable between the DMSO (as a control)-instilled and STF-instilled mice both LIM and non-LIM eyes (Fig. 3c, d and Supplemental Fig. 6c). All the above 3 parameters showed myopic changes when compared to NL and -30D eyes in the DMSO and STF groups (Supplemental Fig. 6a, b, d). Unexpectedly, GSK and NFV instillation induced a myopic shift in refraction, axial elongation, and attenuation of VCD + RT shortening in the control, no lens wearing eyes (Fig 3c, d and Supplemental Fig. 6c). As a result, the myopic phenotype of intraocular differences between the control and minus lens-wearing eyes of the GSK- and NFV-treated groups were diminished (Supplemental Fig. 6a, b, d). These results suggested that

myopia is induced by GSK and NFV administration, i.e., inhibition/derangement of the PERK and ATF6 pathways.”

Fig. 3c and d: Comparison of axial length (Fig. 3c) and refraction (Fig. 3d) to inhibitors instillation under same lens conditions (left: no lens eyes, right: lens-wearing eyes).

Supplemental Fig. 6a and b: Comparison of axial length (Fig. 3c) and refraction (Fig. 3d) to minus lens wearing under same inhibitor-treated condition (Right-left comparison in same mouse).

Comment 8:

In Fig 3h, to confirm the result of PERK and ATF6 pharmacochemical intervention, they generated SaCas9-derived PERK and ATF6 knock-down model and tested the myopia development. Although it is impressive approach, they did not validate the model carefully. The error bar of control appears highly variable. Please confirm protein expression levels using western blot assay. Or please provide the other validation data such as DNA, RNA sequencing, or gene set enrichment analysis.

Response: In accordance with this comment, we performed additional AAV administration experiments and Western blotting of the sclera and added the results as Figure 4d.

Fig.4c: **Effect of AAV-mediated gene manipulation using CRISPR/Cas9 system on PERK and ATF6 expression in sclera.** Twenty-eight days after sub-tenon's capsule injection of AAV-DJ packaging SaCas9 or guide RNA. When the appropriate combination of AAVs is administered, the expression of only the corresponding molecules is reduced.

Comment 9:

In Fig 3i and 3j, the genetic data of ATF6 or ATF6/PERK is not consistent with the pharmacochemical data of ATF6 and PERK inhibitors. Please discuss the reason.

Response: We focused on the difference in the amount of 90kDa ATF6 and made the following observations, assuming that ATF6 has a function other than as a transcription factor.

Under ER stress condition, 90-kDa form of ATF6 (ATF6-P) translocates to the Golgi, where it is cleaved by Site-2 proteinase (S2P), liberating 50-kDa N-terminal fragment (ATF6-N) can serve as transcriptional factor in nucleus. Regarding the involvement of ATF6 in myopia, there was some discrepancy between the results using inhibitors and the results of gene knockdown using the CRISPR/Cas9 system. Ocular administration of NFV or CeapinA-7 resulted in a myopic phenotype in non-myopia-induced eyes, and the difference from myopia-induced eyes disappeared. On the other hand, ATF6 knockdown showed no myopia in either non-myopia-induced or myopia-induced eyes. The two groups differed in the phenotype of the non-myopia induced eye but showed the same phenotype in that the left-right difference disappeared. Ceapin A-7 prevents activation of the ATF6 pathway by retaining ATF6 in the ER, i.e., by inhibiting its transport to the Golgi, thereby preventing its active form (cleaved ATF6) generation. NFV, an HIV protease inhibitor, also inhibits site-2 protease which is a key component of the cleavage (activation) of ATF6 in Golgi apparatus. Both inhibitors suppress the activity of ATF6 by preventing it from being cleaved, resulting in the accumulation of the uncleaved form (ATF6-P) as shown in Fig 3a and b. On the other hand, knockdown of the ATF6 gene using CRISPR/Cas9 system reduces its gene expression, which in turn decreases the expression level of ATF6-P, thereby attenuating the ATF6 pathway. Both interventions on ATF6 reduce the amount of ATF6-N under ER stress, but they work in opposite ways on the amount of ATF6-P. This difference in effect may have led to the difference in the results of this study, i.e., the increase in ATF6-P by drug treatment induced myopia due to its unknown function, while the decrease in ATF6-N may have suppressed the induction of myopia by negative lens wearing.

Comment 10:

In Fig 3k, please explain incubation time of drug instillation in figure legend. And please update p-eIF2, ATF6-P blots which are hard to see the difference.

Response: In accordance with this comment, we added the information about incubation time in figure legend and update eIF2 and ATF6 blots as below. One hour after administration of CCT or AA drops, sclera were collected for Western blotting.

“One hour after eye drop, the sclerae were harvested and subjected to Western blotting.”

Fig.5a: Effect of CCT020312 or AA147 instillation on IRE1m eIF2 and ATF6 activation in sclera.

One hour after administration of CCT or AA drops, sclera were collected for Western blotting. CCT activated only eIF2, while AA activated only the ATF6 pathway.

REVIEWERS' COMMENTS

Reviewer #1 (Remarks to the Author):

The authors responded adequately and extensive to the point raised. I have no extra comments.

Reviewer #2 (Remarks to the Author):

The authors have addressed our comments.

Reviewer #3 (Remarks to the Author):

The authors have carefully updated the data of that the pathway of PERK and ATF6 control axial elongation of myopia pathology. Overall, they have answered the reviewer's comments.

There are a few minor mistakes and missing control data.

1. Line 441~451: please add detail information of catalog number.
2. Line 581: font error.for actin. Please update it.
3. Figure 1d: add molecular weight markers for the immunoblots and full-length uncropped blots
4. Figure 2a, 2e, 2i: add molecular weight markers for the immunoblots and full-length uncropped blots
5. Figure 2b: typo on figure b, y axis "toal">total. Please update it.
6. Figure 2 e: typo on figure "eyd drop">eye drop. Please update it.
7. Figure 3a, 3e: add molecular weight markers for the immunoblots and full-length uncropped blots
8. Figure 4c: add molecular weight markers for the immunoblots and full-length uncropped blots
9. Figure 5a: add molecular weight markers for the immunoblots and full-length uncropped blots
10. Figure 6 c: typo on figure "eyd drop">eye drop. Please update it.
11. Figure 6c: add molecular weight markers for the immunoblots and full-length uncropped blots
12. Supplemental Figure 1h: typo on y axis "normlized">normalized. Please update it.
13. Supplemental Figure 2b: typo -on y axis, "Fluorescence" > Fluorescence. Please update it.
14. Supplemental Figure 5f and 5k: typon on y axis, "differnce"> difference. Please update it.

Response to Reviewer #1

The authors responded adequately and extensive to the point raised. I have no extra comments.

- We wish to express our strong appreciation to you for your insightful comments on our paper. We feel the comments have helped us make significant improvements.

Sincerely yours,
Shin-ichi Ikeda
Toshihide Kurihara
Kazuo Tsubota

Response to Reviewer #2

The authors have addressed our comments.

- We would like to thank you sincerely for your constructive and thought-provoking advice, which has helped us to improve our research of view. We appreciated your kind support.

Sincerely yours,
Shin-ichi Ikeda
Toshihide Kurihara
Kazuo Tsubota

Response to Reviewer #3

The authors have carefully updated the data of that the pathway of PERK and ATF6 control axial elongation of myopia pathology. Overall, they have answered the reviewer's comments. There are a few minor mistakes and missing control data.

- We wish to express our strong appreciation to you for your insightful comments on our paper. Based on your additional comments, we have revised our manuscript as follows:

Comment 1:

Line 441~451: please add detail information of catalog number.

Response: In accordance with this comment, we have added catalog number.

Comment 2:

Line 581: font error.for actin. Please update it.

Response: In accordance with this comment, we revised it.

Comment 3:

Figure 1d: add molecular weight markers for the immunoblots and full-length uncropped blots

Response: In accordance with this comment, we have added molecular weight markers. And all uncropped blots were included in the source data file.

Comment 4:

Figure 2a, 2e, 2i: add molecular weight markers for the immunoblots and full-length uncropped blots

Response: As mentioned in response to Comment 3, we have added molecular weight markers. And all uncropped blots were included in the source data file.

Comment 5:

Figure 2b: typo on figure b, y axis "toal">total. Please update it.

Response: In accordance with this comment, we revised it.

Comment 6:

Figure 2 e: typo on figure "eyd drop">eye drop. Please update it.

Response: In accordance with this comment, we revised it.

Comment 7:

Figure 3a, 3e: add molecular weight markers for the immunoblots and full-length uncropped blots

Response: As mentioned in response to Comment 3, we have added molecular weight markers. And all uncropped blots were included in the source data file.

Comment 8:

Figure 4c: add molecular weight markers for the immunoblots and full-length uncropped blots

Response: As mentioned in response to Comment 3, we have added molecular weight markers. And all uncropped blots were included in the source data file.

Comment 9:

Figure 5a: add molecular weight markers for the immunoblots and full-length uncropped blots

Response: As mentioned in response to Comment 3, we have added molecular weight markers. And all uncropped blots were included in the source data file.

Comment 10:

Figure 6 c: typo on figure “eyd drop”>eye drop. Please update it.

Response: In accordance with this comment, we revised it.

Comment 11:

Figure 6c: add molecular weight markers for the immunoblots and full-length uncropped blots

Response: In accordance with this comment, we have added molecular weight markers. And all uncropped blots were included in the source data file.

Comment 12:

Supplemental Figure 1h: typo on y axis “normlized”>normalized. Please update it.

Response: In accordance with this comment, we revised it.

Comment 13:

Supplemental Figure 2b: typo -on y axis, “Fluorescence” > Fluorescence. Please update it.

Response: In accordance with this comment, we revised it.

Comment 14:

Supplemental Figure 5f and 5k: typon on y axis, “differnce”> difference. Please update it.

Response: In accordance with this comment, we revised it.